# Neuroinflammation—A Crucial Factor in the Pathophysiology of Depression—A Comprehensive Review

**DOI:** 10.3390/biom15040502

**Published:** 2025-03-30

**Authors:** Andreea Sălcudean, Cristina-Raluca Bodo, Ramona-Amina Popovici, Maria-Melania Cozma, Mariana Păcurar, Ramona-Elena Crăciun, Andrada-Ioana Crisan, Virgil-Radu Enatescu, Ileana Marinescu, Dora-Mihaela Cimpian, Andreea-Georgiana Nan, Andreea-Bianca Sasu, Ramona-Camelia Anculia, Elena-Gabriela Strete

**Affiliations:** 1Department of Ethics and Social Sciences, George Emil Palade University of Medicine, Pharmacy, Science and Technology of Targu Mures, 540142 Târgu Mureș, Romania; andreea.salcudean@umfst.ro (A.S.); melaniacozma76@gmail.com (M.-M.C.); dora_cimpian@yahoo.com (D.-M.C.); 2Department of Management and Communication in Dental Medicine, Faculty of Dental Medicine, Victor Babes University of Medicine and Pharmacy of Timisoara, 9 Revolutiei 1989 Bv., 300070 Timisoara, Romania; 3Orthodontic Department, Faculty of Dental Medicine, George Emil Palade University of Medicine, Pharmacy, Science and Technology of Targu Mures, 540142 Târgu Mures, Romania; marianapac@yahoo.com; 4Association “Child and Childhood”, 540472 Targu Mures, Romania; r.craciun@yahoo.com; 5Doctoral School, George Emil Palade University of Medicine, Pharmacy, Science and Technology of Targu Mures, 540142 Târgu Mureș, Romania; crisanandrada@yahoo.com; 6Department of Psychiatry, Faculty of Medicine, Victor Babes University of Medicine and Pharmacy of Timisoara, 300041 Timisoara, Romania; enatescu.virgil@umft.ro; 7Discipline of Psychiatry, Faculty of Medicine, University of Medicine and Pharmacy of Craiova, 200349 Craiova, Romania; ileana.marinescu@umfcv.ro; 8First Department of Psychiatry, Clinical County Hospital of Targu Mures, 540142 Târgu Mureș, Romania; nandree96@yahoo.com (A.-G.N.); andreea_vinteler@yahoo.com (A.-B.S.); 9Discipline of Occupational Medicine, Faculty of Medicine, Victor Babes University of Medicine and Pharmacy of Timisoara, 300041 Timișoara, Romania; ramona.anculia@umft.ro; 10Department of Psychiatry, George Emil Palade University of Medicine, Pharmacy, Science and Technology of Targu Mures, 540142 Târgu Mureș, Romania; elena.buicu@umfst.ro

**Keywords:** neuroinflammation, major depressive disorder, cytokines, microglia, astrocytes, neurotransmitters, neurotoxicity, hypothalamic–pituitary–adrenal axis

## Abstract

Depression is a multifactorial psychiatric condition with complex pathophysiology, increasingly linked to neuroinflammatory processes. The present review explores the role of neuroinflammation in depression, focusing on glial cell activation, cytokine signaling, blood–brain barrier dysfunction, and disruptions in neurotransmitter systems. The article highlights how inflammatory mediators influence brain regions implicated in mood regulation, such as the hippocampus, amygdala, and prefrontal cortex. The review further discusses the involvement of the hypothalamic–pituitary–adrenal (HPA) axis, oxidative stress, and the kynurenine pathway, providing mechanistic insights into how chronic inflammation may underlie emotional and cognitive symptoms of depression. The bidirectional relationship between inflammation and depressive symptoms is emphasized, along with the role of peripheral immune responses and systemic stress. By integrating molecular, cellular, and neuroendocrine perspectives, this review supports the growing field of immunopsychiatry and lays the foundation for novel diagnostic biomarkers and anti-inflammatory treatment approaches in depression. Further research in this field holds promise for developing more effective and personalized interventions for individuals suffering from depression.

## 1. Introduction

Neuroinflammation is defined as the inflammatory response within the central nervous system (CNS), mediated primarily by glial cells such as microglia and astrocytes. Neuroinflammation plays a key role in the etiology of major depressive disorder (MDD) [1,2]. Increased levels of proinflammatory cytokines have been consistently observed in people with depression [2,3]. These cytokines can disrupt neurotransmitter systems, in particular the serotonergic and dopaminergic pathways, which are crucial for mood regulation [4,5].

The activation of microglia in response to stress or injury leads to the release of inflammatory mediators that can alter neurotransmitter synthesis and receptor sensitivity [6,7,8]. For example, increased expression of indoleamine 2,3-dioxygenase (IDO) during inflammatory states can lead to depletion of tryptophan, a precursor for serotonin, thereby contributing to serotonergic dysfunction in depression [9]. In addition, the kynurenine pathway, activated by inflammatory cytokines, diverts tryptophan metabolism away from serotonin production, leading to increased levels of neuroactive metabolites that may have neurotoxic effects [10]. Microglial activation is a hallmark of neuroinflammation and has been shown to significantly influence neurotransmitter systems. Activated microglia release various cytokines and chemokines that can modulate neuronal function and neurotransmitter release [7,11]. For example, the release of IL-1β and TNF-α can inhibit the synthesis of key neurotransmitters, leading to an imbalance associated with mood regulation [12,13]. In addition, microglial activation may lead to alterations in glutamatergic neurotransmitters, which are involved in the pathophysiology of depression [7,9].

Research has shown that chronic stress can exacerbate microglial activation, further perpetuating neuroinflammation and neurotransmitter dysregulation [14]. In animal models, chronic unpredictable mild stress (uCMS) has been shown to increase microglial activation and levels of pro-inflammatory cytokines, correlating with depressive-like behaviors and altered neurotransmitter levels [15,16]. This suggests that the interplay between stress, neuroinflammation, and neurotransmitter systems is a critical area of investigation in understanding depression. In addition, exploring neuroinflammation in the context of depression aligns with the growing field of immunopsychiatry, which seeks to understand the interplay between the immune system and mental health [17]. This interdisciplinary approach not only broadens the scope of research but also promotes collaboration between psychiatry, neurology, and immuno-technology, leading to innovative therapeutic strategies. As the field continues to evolve, it is essential to consider the implications of neuroinflammation in the development of personalized medicine approaches to depression, where treatments can be tailored based on individual inflammatory profiles [18,19]. The complex relationship between neuroinflammation and depression highlights the need for a comprehensive understanding of both conditions. By investigating the biological underpinnings of neuroinflammation and its impact on mood disorders, researchers can help develop interventions that are more effective and improve the quality of life of people affected by depression.

Depression is associated with significant alterations in different brain regions, affecting both functional connectivity and structural integrity. The main regions involved in depression include the hippocampus, prefrontal cortex, amygdala, anterior cingulate cortex (ACC), insular cortex, and nucleus accumbens. It has been shown that people with depression often have volumetric reductions in the hippocampus. Furthermore, neuroimaging studies have shown significant decreases in hippocampal volume among depressed patients, correlating these anatomical changes with the severity of depressive symptoms and cognitive dysfunction [20,21]. The hippocampus is particularly sensitive to stress-related neurochemical changes, which may contribute to its atrophy in depression [22]. Specifically, the dorsolateral prefrontal cortex and ventromedial sectors showed reductions in gray matter volume and changes in connectivity in depressed subjects [23,24]. These changes are thought to contribute to the deficits in executive functioning and emotional control that characterize MDD [25,26]. In addition, reduced activity in the prefrontal cortex was associated with increased emotional reactivity in the amygdala, emphasizing its regulatory role [27,28]. The amygdala is overactive in people with depression, which may correlate with heightened emotional responses and negative affective states [23,28]. Structural and functional changes in ACC are predominant in people with depression [29,30]. During emotional states, increased ACC activation and decreased volume have been frequently reported, linking these changes to the severity and type of depressive symptoms [27,30]. In addition, functional connectivity analyses indicate that ACC may exhibit disrupted connectivity with other key brain regions involved in mood regulation, suggesting its role in the network dysfunction associated with depression [29,30]. The insular cortex is involved in interoceptive awareness and emotional experience. Changes in the structural and functional properties of the insula have been identified in people suffering from depression, affecting their role in emotions and self-awareness. Dysfunction in this area has previously been associated with impaired sensitivity to affective states and emotional processing, further complicating depression [22]. The nucleus accumbens is a key region in the brain’s reward circuitry and has been implicated in the motivational aspect of mood disorders. Alterations in the volume and functional connectivity of the nucleus accumbens are linked to anhedonia and decreased motivation, core symptoms of depression [20,28]. Dysregulation within this circuit may contribute to the failure to experience pleasure commonly seen in depressed people [23,30].

This review article aims to provide a comprehensive synthesis of the current evidence regarding the role of neuroinflammation in the pathophysiology of depression. By exploring the cellular, molecular, and systemic mechanisms underlying neuroinflammatory processes, particularly glial cell activation, cytokine signaling, HPA axis dysregulation, and peripheral immune system involvement, this review seeks to elucidate how these pathways contribute to mood dysregulation and depressive symptoms. Furthermore, the article highlights the bidirectional relationship between depression and inflammation, the impact on neurotransmitter systems, and the relevance of neuroimmune crosstalk. Through this integrative approach, the review aims to support the development of novel diagnostic biomarkers and inform future therapeutic strategies targeting inflammatory pathways in depression.

## 2. Neuroinflammation and Depression: A Complex Relationship

### 2.1. Neuroinflammation: Cellular and Molecular Aspects

Neuroinflammation is increasingly recognized as a central feature in the pathogenesis of various neurodegenerative diseases, including Alzheimer’s disease (AD) and Parkinson’s disease (PD). This complex biological process involves multiple cellular and molecular mechanisms, mainly characterized by the activation of glial cells, especially microglia, which are the resident immune cells of the CNS [31,32]. The role of microglia in neuroinflammation is particularly significant because they are the main immune response in the CNS. Microglia can adopt different states of activation, becoming potentially neuroprotective or neurotoxic depending on the context of their activation. In diseases such as prion disorders, it has been shown that microglial activation may contribute to disease progression through the release of proinflammatory mediators [33,34]. Specifically, several studies indicate that microglia and other glial cells work in tandem, whereby neuroinflammation can amplify neuronal cell death, leading to a cascade effect that exacerbates neurodegeneration [35,36]. In general, microglia protect neuronal integrity by clearing cellular debris and modulating inflammatory responses. However, when activated by various pathological stimuli, including neurodegeneration, microglia can adopt a proinflammatory phenotype, leading to the release of cytokines, chemokines, and other mediators that perpetuate inflammation and contribute to neuronal loss [37,38].

In AD, neuroinflammation is closely linked to the presence of amyloid-beta plaques and neurofibrillary tangles, which trigger microglial activation. Increased levels of inflammatory mediators have been correlated with cognitive decline in AD patients, indicating that persistent neuroinflammation may exacerbate neurodegenerative processes [39,40]. Recent studies have shown that activated microglia not only mediate inflammatory responses but also participate in neurotoxic pathways that can lead to neuronal apoptosis [40,41]. Targeting neuroinflammatory pathways may offer a promising therapeutic pathway to alleviate the cognitive deficits associated with AD [40].

Similarly, in PD, chronic neuroinflammation is observed as a key contributor to the degeneration of dopaminergic neurons [42,43]. The activation of microglia in this context is often exacerbated by the presence of endogenic toxins released by damaged neurons, leading to a vicious cycle of neuroinflammation and neurodegeneration [44]. In addition, neuroinflammatory processes in PD are characterized by increased production of proinflammatory cytokines, which may promote further neuronal damage [45].

The role of high mobility group box 1 (HMGB1) as a pro-inflammatory mediator has attracted significant attention in recent years. HMGB1 can activate microglia via the Toll-like receptor 4 (TLR4), which enhances the inflammatory response and contributes to the development of depressive behaviors [46,47]. In addition, the interaction of HMGB1 with neuronal cells may lead to neurotoxicity, thus linking neuroinflammation to the neurodegeneration observed in various psychiatric disorders [47]. The targeting potential of HMGB1 in therapeutic strategies demonstrates the translational importance of these molecular factors in combating depression.

In addition to microglial activation and cytokine release, astrocytes and T-helper (Th) cells have been implicated in depression-related neuroinflammatory processes. Astrocytes, when activated, can exacerbate neuroinflammation by secreting several proinflammatory factors, contributing to a vicious cycle of inflammation and depressive symptoms [48]. In addition, Th17 cells have recently been recognized as crucial in stimulating non-neo-inflammation related to neuropsychiatric disorders, including multiple sclerosis (MS) and depression, exemplifying the influence of the immune system on mental health [49].

A feature of neuroinflammatory responses includes the expression and activation of the Nod-like receptor protein 3 (NLRP3) inflammasome, particularly in the context of AD and PD. The NLRP3 inflammasome has been implicated in the processing and secretion of pro-inflammatory cytokines, such as IL-1β, contributing to the chronic inflammatory state observed in these disorders [40,50,51]. The activation of this multiprotein complex relies heavily on generated signals from pathogen-associated molecular patterns (PAMPs) or damage-associated molecular patterns (DAMPs), emphasizing the critical interplay between innate and adaptive immune responses in the brain [42,52]. It has been suggested that inhibition of the NLRP3 pathway could significantly reduce depressive-like behaviors [53].

The temporal dynamics of neuroinflammation also warrant attention, as evidence suggests that microglial activation at an early stage may initially confer neuroprotective effects, while prolonged activation leads to deleterious consequences [32,54]. This is exemplified in conditions such as traumatic brain injury (TBI), in which initial inflammatory responses are followed by a chronic phase that exacerbates neuronal damage and impairs recovery [55,56].

### 2.2. Peripheral and Central Immune System Interactions in Depression

The interaction between the peripheral immune system and the CNS plays a complex role in the pathophysiology of depression. The research demonstrates that both systems engage in bidirectional communication, where inflammatory mediators in the immune response can influence neuronal pathways and vice versa.

One of the key observations linking immune dysfunction to depression is the role of pro-inflammatory cytokines. Studies have shown that individuals with MDD have elevated levels of these cytokines in both peripheral blood and cerebrospinal fluid, indicating systemic inflammation that may penetrate the blood–brain barrier [4,57]. Increased levels of cytokines (e.g., IL-6 and TNF-α) are often correlated with depressive symptoms, supporting the idea that inflammation is an essential component of the etiology of depression [4]. In addition, peripheral immune activation has been shown to provoke neuroinflammatory responses in the brain, leading to changes in neurotransmitter systems essential for mood regulation [58].

Microglia serve as a crucial interface between the immune system and the pathology of depression. Activated microglia release pro-inflammatory cytokines, which can negatively affect neuronal function and neuroplasticity [59]. This neuroinflammation contributes to the loss of synaptic connectivity and neurogenesis, processes necessary for mood stability [6,60]. In addition, the interaction of microglial activity with psychological and physiological stressors highlights the potential for environmental influences to exacerbate neuroinflammation, thereby increasing vulnerability to depressive states [61,62].

Stress-related neuroimmune modulation is a phenomenon that deserves attention in the study of depression. Exposure to chronic stress has been shown to activate the peripheral immune response, leading to an increase in circulating cytokines that subsequently activate microglia [61,63]. This cascade not only affects neurotransmission but can also induce structural changes in the brain, particularly in regions such as the hippocampus, which plays a significant role in emotional regulation and cognition [64]. Accordingly, stress-induced neuroinflammation may catalyze the onset of MDD, particularly in susceptible individuals who have experienced significant stressors or trauma [65].

Genetic predisposition also emerges as a significant factor influencing immune system interactions in depressive disorders. One study has highlighted genetic factors associated with immune responses that may predispose people to depression, particularly in the context of childhood maltreatment and subsequent inflammatory responses in adulthood [66]. These findings emphasize the complex interplay between genetic susceptibility, environmental stressors, and immune activation pathways in the etiology of MDD.

### 2.3. Bidirectional Relationship: Can Depression Itself Trigger Neuroinflammation?

The bidirectional relationship between depression and neuroinflammation is an increasingly important topic in current psychological and neurological research. This relationship suggests that not only depression may lead to neuroinflammatory processes, but also neuroinflammation, in turn, may contribute to the manifestation and exacerbation of depressive symptoms. The role of progranulins derived from various gliocytes in the nucleus accumbens (NAc) has been highlighted, suggesting that neuroinflammation in this region is critically related to depression-like behaviors. Experimental evidence confirms that the induction of neuroinflammation in the NAc by immune challenges can lead to depressive-like behaviors in animal models, underscoring the importance of this brain region in mood regulation and pathogenesis of depression [64]. NAc is a key component of the brain’s reward circuitry and is involved in regulating mood, motivation, and pleasure. Dysregulation of neuroinflammatory processes in the NAc can lead to significant changes in neurotransmitter systems, neuroplasticity, and ultimately depression. Proinflammatory cytokines may influence NAc function by altering dopaminergic and glutamatergic neurotransmitter activity. Guo et al. demonstrated that microRNA-15b affects synaptic protein levels and function in the NAc, leading to depression-like behavior in mice [67]. Moreover, chronic stress induces neuroinflammation in the NAc. The stress of chronic social defeat leads to increased levels of proinflammatory cytokines in the NAc, which may impair dopaminergic signaling and contribute to anhedonia, a core symptom of depression [68]. Microglia activation in the NAc following exposure to stress may further exacerbate neuroinflammation. The relationship between neuroinflammation and neurotransmitter dysregulation in the NAc is particularly relevant in the context of dopamine signaling. The NAc receives dopaminergic input from the ventral tegmental area (VTA), and alterations in this pathway can have a significant impact on reward processing and motivation. Kushnareva et al. found that neurobiological abnormalities in the NAc were associated with depressive symptoms in a rat model of the anxiety-depressive state, indicating the importance of this region in mood disorders [69].

Specific molecular pathways have been implicated in the relationship between depression and neuroinflammation. The TLR4 signaling pathway, known for its role in mediating the immune response, has been linked to stress-related neuroinflammation and depression. Research shows that stress-triggered neuroinflammation increases TLR4 receptor expression, indicating a mechanistic pathway by which stress-induced inflammation may facilitate depressive behavior [57]. This relationship suggests that therapies targeting neuroinflammatory processes could serve as potential interventions for alleviating depressive symptoms, while also highlighting the multifaceted nature of depression as being influenced by both psychological stress and inflammation [70].

Specific neuroinflammatory processes reveal additional connections. Activated microglia release pro-inflammatory cytokines, which may contribute to synaptic dysfunction and neurodegeneration, processes often observed in people with MDD [71]. Chronic neuroinflammation not only affects neurogenesis but also influences the neuroendocrine system by disrupting the HPA axis, which is frequently associated with depression. This dysregulation may perpetuate a cycle in which the neuroinflammation exacerbates the symptoms of depression, complicating treatment efforts [72].

Peripheral immune responses and neuroinflammatory markers such as translocator protein (TSPO) correlate significantly with depressive symptoms and behaviors. Increased TSPO density has been observed during depressive episodes, providing insight into how neuroinflammation may manifest as a mood disorder, thus reinforcing the need for further research on neuroinflammatory biomarkers in a clinical setting [73].

Underlying the complex interplay between depression and neuroinflammation is a bidirectional relationship in which depression can induce a neuroinflammatory state, leading to impairments in brain function that can further exacerbate mood disorders. Chronic psychological stress not only activates neuroinflammatory pathways but also affects neurotransmitter systems essential for mood regulation [61]. In addition, inflammatory markers are consistently found in patients with treatment-resistant depression, indicating that unresolved inflammatory responses may be a significant factor in the persistence of depressive symptoms [74].

## 3. Biological Mechanisms Underlying Neuroinflammation in Depression

### 3.1. The HPA Axis and Stress

Activation of the HPA axis is a key biological mechanism linking stress responses to depression. The HPA axis is a complex neuroendocrine system that regulates the body’s response to stress by releasing hormones, mainly cortisol. When an individual is faced with a stressor, the hypothalamus releases corticotropin-releasing hormone (CRH), which stimulates the anterior pituitary gland to secrete adrenocorticotropic hormone (ACTH), which stimulates the adrenal glands to produce cortisol, which plays a key role in the stress response [75,76]. Research has shown that hyperactivity of the HPA axis is associated with the onset and persistence of depressive symptoms. For example, Zajkowska et al. conducted a systematic review that found elevated cortisol levels in adolescents and young adults with depression, suggesting that HPA axis hyperactivity serves as a marker for stress and a precursor to depressive disorders [77]. This finding aligns with studies in adults, which also indicate that chronic stress and high cortisol levels are linked to the development of major depressive disorder [78,79].

The relationship between HPA axis activation and neuroinflammation is particularly significant. Chronic stress and the resulting hyperactivation of the HPA axis can lead to increased production of pro-inflammatory cytokines, which are known to disrupt neurotransmitter systems, impair neuroplasticity, and ultimately lead to mood dysregulation [80]. For example, Benatti et al. emphasized that the interplay between HPA axis hyperactivity and neuroinflammation is crucial in the etiopathogenesis of MDD, suggesting that targeting these pathways may enhance the efficacy of antidepressant treatments [81]. In addition, activation of the HPA axis can create a feedback loop that perpetuates neuroinflammation. The cytokines released during inflammatory responses may act on the hypothalamus, promoting additional release of CRH and ACTH, which exacerbates HPA axis activation [82]. This cyclical relationship between HPA axis dysregulation and neuroinflammation highlights the complexity of the biological mechanisms underlying depression. The dysregulation of this axis is closely linked to alterations in various neurotransmitter systems, including serotonin, norepinephrine, and dopamine, which are essential in mood regulation and the pathophysiology of depression [1,83,84]. One of the most important neurotransmitter systems affected by the activation of the HPA axis is the serotonergic system. Chronic stress and the resulting hyperactivity of the HPA axis have been shown to lead to a decrease in serotonin levels in the brain, particularly in regions such as the prefrontal cortex (PFC) and hippocampus, which are essential for mood regulation. The serotonin transporter (SERT) and serotonin receptors, in particular the 5-HT1A and 5-HT2A receptors, are also involved in this process. Deregulation of these re-receptors can lead to altered serotonergic signaling, contributing to depressive symptoms. In addition, the interaction between glucocorticoid and serotonin receptors has been shown to modulate the effects of stress on mood, indicating a complex interplay between these systems [1,84].

The norepinephrine system is similarly affected by activation of the HPA axis. Norepinephrine, produced mainly in the locus coeruleus, is important for the body’s response to stress. Chronic activation of the HPA axis can lead to increased levels of norepinephrine, which may initially serve to increase alertness but can eventually contribute to anxiety and depressive symptoms when deregulated. Alpha-2 adrenergic receptors, which play a role in inhibiting the release of norepinephrine, may become desensitized due to prolonged stress, leading to a further exacerbation of depressive symptoms [1,83]. Dopamine, another critical neurotransmitter involved in the reward system, is also significantly affected by HPA axis dysregulation. Chronic exposure to high levels of cortisol can lead to a reduction in dopamine synthesis and receptor sensitivity, particularly in the mesolimbic pathway, which is associated with reward processing and motivation. This reduction in dopaminergic signaling can manifest as anhedonia, a core symptom of depression characterized by a diminished ability to experience pleasure. The interaction between glucocorticoids and dopamine receptors, particularly the D2 receptor, further complicates the relationship between HPA axis activation and depressive symptoms [85]. In addition to these classical neurotransmitter systems, the role of vasopressin and neuropeptide Y (NPY) neuropeptides in the context of HPA axis activation and depression is gaining attention. Vasopressin, which is released together with CRH during stress responses, has been implicated in the modulation of anxiety and depressive behaviors. The vasopressin V1B receptor, in particular, has been identified as a potential target for antidepressant therapies, as its antagonism relieves HPA axis dysfunction and improves mood. Conversely, NPY has been shown to exert anxiolytic effects and counteract the effects of stress on the HPA axis, exerting a protective role against stress-induced mood disorders [86,87].

The impact of HPA axis activation on depression is not uniform and may vary depending on individual factors such as genetics, early life stress, and environmental influences. For example, research by Booij et al. has shown that the chronicity of depressive problems influences HPA axis responses to psychosocial stress, with recent-onset depressive problems predicting increased HPA axis reactivity. This suggests that the timing and context of stress exposure may shape the HPA axis response and its subsequent impact on mood [88]. Furthermore, the role of glucocorticoid receptors (GRs) in mediating the effects of cortisol is crucial in understanding HPA axis activation in depression. Chronic stress may lead to a reduction in GR expression, affecting the negative feedback mechanism that normally regulates cortisol levels [89]. This dysregulation can lead to a state of glucocorticoid resistance, in which the body becomes less responsive to cortisol, further perpetuating the cycle of stress and depression [90].

The effects of HPA axis activation extend beyond mood symptoms; they can also affect cognitive function. High cortisol levels have been associated with memory and learning deficits, which are often seen in people with depression [91]. This cognitive dysfunction is mediated by the effects of cortisol on hippocampal neurogenesis and synaptic plasticity, both of which are essential for cognitive health [92].

### 3.2. Cytokines as Mediators of Depression

Proinflammatory cytokines have attracted significant attention in recent years for their role as mediators of depression. These signaling molecules are produced by various immune cells in response to stress, infection, and injury and play a crucial role in the inflammatory response. In the context of depression, an imbalance in proinflammatory cytokines may contribute to the development and persistence of depression. Among the variety of pro-inflammatory cytokines released in neuroinflammation, TNF-α, IL-1β, and IL-6 are frequently cited in the literature as being elevated in MDD patients [93,94,95]. However, a more nuanced understanding shows that other cytokines, such as interleukin-10 (IL-10) and interferon-gamma (IFN-γ), also play significant roles in the neuroinflammatory processes associated with depression [96,97,98].

Figure 1 shows two stages of neuroinflammatory processes in the brain in response to stress: (a) acute stress and (b) chronic stress or depression. It illustrates the role of microglia and astrocytes, the glial cells involved in neuroinflammatory responses, and highlights the transition from controlled inflammation during acute stress to sustained and damaging inflammation during chronic stress or depression. The acute stress response triggers the release of pro-inflammatory cytokines (e.g., TNF-α, IL-1β, IL-6) from peripheral tissues into the bloodstream. These molecules act as signals, indicating the presence of stress in the brain. Upon detection of these cytokines, microglia and astrocytes become activated. This activation is part of the brain’s response to maintain homeostasis and adapt to stress. During acute stress, microglia and astrocyte activation is regulated to avoid excessive inflammation. This controlled inflammatory response is protective and helps the brain cope with the stress factor without damaging neural tissues. In chronic stress or depression, there is a permanent release of pro-inflammatory cytokines from peripheral tissues, similar to acute stress but sustained over time. Continued exposure to proinflammatory cytokines leads to prolonged and excessive activation of microglia and astrocytes. Unlike the controlled response seen in acute stress, this prolonged activation is unregulated and leads to a proinflammatory environment in the brain. Chronic activation of these glial cells causes a continuous inflammatory response, which contributes to neuronal damage. Increased inflammation is associated with reduced levels of neurotransmitters such as serotonin (5-HT) and dopamine, increased HPA axis activity, and higher levels of glutamate, all of which affect brain plasticity and contribute to symptoms of depression (Figure 1).

One of the most studied proinflammatory cytokines related to depression is IL-6. It is produced by various cell types, including astrocytes and microglia, in response to inflammatory stimuli. Elevated levels of IL-6 have been documented in patients with depression and this cytokine is known to influence the HPA axis, which plays a significant role in the stress response and mood regulation [6]. HPA axis dysregulation due to increased IL-6 can lead to changes in cortisol levels, further exacerbating depressive symptoms [12]. The inflammatory hypothesis of depression hypothesizes that elevated IL-6 levels contribute to the pathophysiology of depression by disrupting neurotransmitter systems and promoting neuroinflammation [99]. In addition, IL-6 has been shown to affect neurotransmitter systems by modulating the expression of serotonin receptors, thereby influencing mood and behavior [100,101].

Another key proinflammatory cytokine involved in depression is TNF-α, produced by activated microglia, influencing neuronal function and survival. Increased levels of TNF-α have been observed in both peripheral blood and cerebrospinal fluid of depressed patients [102]. This cytokine can induce apoptosis in neurons and glial cells, leading to neurodegeneration and cognitive deficits [6]. Tsamakis et al. reported that TNF-α can activate CNS circuitry, leading to adverse neurobehavioral responses, including depressed mood and cognitive dysfunction [99]. This highlights the role of TNF-α in mediating the effects of neuroinflammation on mood regulation. In addition, TNF-α has been shown to inhibit the synthesis of BDNF, which is also vital for mood regulation [12]. Inhibition of BDNF by TNF-α may contribute to the pathophysiology of depression, as reduced levels of BDNF are commonly observed in depressed individuals [103].

IL-1β is another pro-inflammatory cytokine that has been implicated in the development of depression and is one of the most studied cytokines. IL-1β can cross the BBB and activate the HPA axis, leading to increased cortisol production and further exacerbating neuroinflammation. Not only does this cytokine promote neuroinflammation but it also disrupts the balance of neurotransmitters, particularly serotonin and dopamine [12]. The activation of IL-1β signaling pathways may disrupt neurotransmitter systems, impair neuroplasticity, and contribute to the cognitive deficits often observed in people with depression. Increased levels of IL-1β have been observed in people with MDD and are associated with increased microglial activation in the brain [95]. In addition, studies have shown that IL-1β can induce IDO expression, leading to reduced serotonin availability and contributing to depressive symptoms [103].

IL-10, often considered an anti-inflammatory cytokine, has been shown to play a dual role in depression. While it may attenuate inflammation, its levels may also be down-regulated in chronic inflammatory states, potentially leading to an inappropriate response to neuroinflammation [104]. Conversely, IFN-γ, a proinflammatory cytokine produced mainly by T cells, has been implicated in exacerbating depressive symptoms through its effects on the kynurenine pathway, which is essential for the metabolization of tryptophan to neuroactive compounds [105,106]. Increased levels of IFN-γ have been observed in patients with depression, suggesting a link between immune activation and mood disorders [73,107]. In addition, IL-12, which is involved in T-cell differentiation, was associated with increased neuroinflammation and depressive symptoms. Studies indicate that elevated IL-12 levels correlate with the severity of depressive symptoms, emphasizing its potential role as a biomarker for depression [97,108].

IL-18 is another cytokine involved in the pathophysiology of depression. It is known to promote the production of proinflammatory cytokines and may influence neurotransmitter systems, in particular serotonin. Increased levels of IL-18 have been observed in patients with MDD, suggesting a link between this cytokine and mood dysregulation [109]. The interaction between IL-18 and the kynurenine pathway, which is involved in tryptophan metabolism, may also contribute to serotonin deficiency, a feature of depression [110].

IL-33, a member of the IL-1 family of cytokines, is also important in the neuroinflammatory landscape associated with depression. It is predominantly expressed in glial cells and astrocytes, and its levels are elevated in various neuropsychiatric disorders, including MDD [111,112]. IL-33 is known to orchestrate the release of other inflammatory cytokines, thereby amplifying the neuroinflammatory response [112]. Studies have shown that IL-33 may influence depression-related behavioral traits, with knockout models exhibiting impaired social recognition and anxiety-like behaviors [113]. This suggests that IL-33 may not only serve as a marker of neuroinflammation but may also play a direct role in the behavioral manifestations of depression.

The relationship between proinflammatory cytokines and depression is also complicated by their interactions with other biological systems. The kynurenine pathway, which is involved in tryptophan metabolism, may be influenced by pro-inflammatory cytokines. Increased cytokine levels may direct tryptophan metabolism to produce kynurenine, a metabolite associated with neurotoxic effects and depressive symptoms. This suggests that proinflammatory cytokines may contribute not only to neuroinflammation but also to altered neurotransmitter metabolism, further linking inflammation to depression [114].

Figure 2 illustrates how proinflammatory cytokines influence the kynurenine pathway, thereby affecting neurotransmitter metabolism and contributing to neuroinflammation, cognitive dysfunction, and neurodegeneration. Increased proinflammatory cytokines direct tryptophan metabolism towards the production of kynurenine, decreasing serotonin availability and generating neurotoxic metabolites (quinolinic acid). This dual impact, neuroinflammation, and altered neurotransmitter metabolism provide a mechanistic basis for the association between inflammation and mood disorders, particularly depression.

In terms of age-related differences, the research indicates that the cytokine response to neuroinflammation may vary between different age groups. For example, older populations often have higher baseline levels of proinflammatory cytokines, which may contribute to the increased prevalence of depression in this demographic [115]. In contrast, younger individuals may show a different cytokine profile, potentially influenced by developmental factors and stage of neuroimmune maturation [116]. This suggests that while certain cytokines are consistently elevated in depression in all age groups, the overall inflammatory response may differ, necessitating age-specific therapeutic approaches. Older adults with depression have elevated levels of IL-6 and TNF-α compared to their non-depressed homologs, suggesting that age-related neuroinflammatory changes may play a significant role in the onset and persistence of depressive symptoms in older adults [12]. In contrast, younger populations, such as children and adolescents, may show different cytokine profiles, with studies indicating elevated levels of IL-6 and IL-10 in children with attention-deficit/hyperactivity disorder (ADHD), which may overlap with depressive symptoms [100]. In addition, the effects of neuroinflammation on neurotransmitter systems may vary between different age groups. In younger people, neuroinflammation may mainly affect the serotonergic system, leading to changes in mood and behavior. At the same time, in older adults, the impact may extend to multiple neurotransmitter systems, including dopaminergic and glutamatergic pathways, leading to more complex neuropsychiatric manifestations [117]. This age-dependent variability emphasizes the importance of considering developmental factors when examining the relationship between neuroinflammation, cytokines, and depression.

The chronic stress model of depression has been instrumental in elucidating the role of cytokines in neuroinflammation. Studies using CUMS have demonstrated that cytokines such as IL-1β and IL-6 are significantly elevated in animal models, correlating with the development of depressive-like behaviors [118,119]. Activation of the NLRP3 inflammasome in microglia has been shown to mediate this response, leading to the release of these cytokines and subsequent neuronal dysfunction [118,120]. Increased levels of IL-1β have been linked to changes in glutamate signaling, which is involved in synaptic plasticity and cognitive function [106,121]. This disruption of glutamate homeostasis can lead to excitotoxicity, further exacerbating depressive symptoms. In addition, the kynurenine pathway, influenced by pro-inflammatory cytokines, has been implicated in the depletion of neuroprotective factors, thus contributing to the neurodegenerative aspects of depression [105].

### 3.3. Microglial and Astrocytic Activation in the Brain

Microglial and astroglial activation in the brain plays an important role in the non-neuroinflammatory processes associated with depression. These glial cells are essential for maintaining homeostasis in the CNS and respond to various pathological stimuli, including stress and injury. Microglia are the resident immune cells of the CNS and act as the first line of defense against injury and infection. Under normal conditions, microglia have a branched morphology, allowing them to continuously monitor their environment. However, upon activation by stressors or injury, microglia undergo morphological changes, becoming amoeboid and proliferating in response to inflammatory signals [122]. This activation is characterized by the release of proinflammatory cytokines, which lead to exacerbation of neuroinflammation [123,124]. In addition, microglia activation can lead to changes in neurotransmitter systems, in particular by modulating receptor expression. It has been argued that microglial activation can lead to increased expression of GABA_B receptors, which are crucial for inhibitory neurotransmission [125]. This is particularly relevant in the context of depression, where deregulation of GABA-ergic signaling has been implicated. In addition, microglia can respond to neurotransmitters such as dopamine and norepinephrine, influencing their activation state and subsequent inflammatory response [126,127]. This interaction suggests a bidirectional communication between neurons and microglia, in which neurotransmitters not only affect neuronal activity but also modulate glial function. Beyond morphological changes, microglia in depression show significant transcriptional deficits. Increased expression of homeostatic markers, such as P2Y12 receptor, TMEM119, CCR5, CD206, CX3CR1, IL-10 receptor, TREM2, and CD68, has been demonstrated. These markers indicate enhanced homeostatic functions of microglia, suggesting a shift towards a protective rather than inflammatory role in the context of MDD. This change may reflect an adaptive response to maintain brain homeostasis amid neurobiological changes associated with depression [128].

The interaction between microglial activation and neurotransmitter systems is also exemplified by the role of specific signaling pathways. Activation of the NF-κB signaling pathway in microglia has been shown to regulate the expression of neurotransmitter receptors and synaptic proteins, highlighting the complex relationship between inflammation and neurotransmission in depression [129]. The impact of neuroinflammation on neurotransmitter systems is further emphasized by the role of specific cytokines [130]. These cytokines can alter the balance between excitatory and inhibitory neurotransmission, leading to the synaptic dysfunction characteristic of depression. The chronic presence of these inflammatory mediators can create a neurotoxic environment that exacerbates glial activation and further impairs neurotransmitter signaling.

Astrocytes, the most abundant glial cells in the brain, also play a crucial role in neuroinflammation. They provide metabolic support to neurons, regulate neurotransmitter levels, and maintain the blood–brain barrier (BBB) [131]. Astrocytes are involved in the uptake and recycling of neurotransmitters, such as glutamate and GABA, thus maintaining synaptic homeostasis. In response to inflammatory stimuli, astrocytes can be activated and may adopt a reactive phenotype, characterized by the increased expression of glial fibrillary acidic protein (GFAP) and release of pro-inflammatory cytokines [132]. Du et al. found that astrocytes contribute significantly to inflammatory responses in the brain, indicating their involvement in the pathogenesis of depression [133]. In depression, astrocytic dysfunction has been observed, leading to impaired neurotransmitter clearance and altered synaptic transmission [128,134]. Astrocyte activation can lead to the production of reactive oxygen species (ROS) and activation of inflammasomes, further perpetuating neuroinflammation [134]. Inflammasomes are multiprotein complexes that play important roles in the innate immune response, mediating the production of pro-inflammatory cytokines, which significantly influence neurotransmitter systems and contribute to the pathophysiology of depression. Psychological stress leads to activation of the NLRP3 inflammasome, resulting in increased levels of IL-1β and other pro-inflammatory cytokines in the brain [1,135]. This inflammatory response disrupts neurotransmitter signaling, particularly the serotonin and dopamine pathways. Increases in NLRP3 components in response to lipopolysaccharide (LPS) exposure further support the idea that neuroinflammation contributes significantly to depressive-like behaviors [135,136]. The NLRP1 inflammasome also plays a role in the neuroinflammatory landscape associated with depression. The research indicates that NLRP1 activation leads to neuroinflammation and has been linked to cognitive impairment and neuronal damage, particularly in the context of aging and neurodegenerative diseases [137,138]. The interplay between oxidative stress and NLRP1 activation is particularly notable, as ROS can trigger activation of the NLRP1 inflammasome, leading to a cascade of inflammatory responses that exacerbate depressive symptoms [137]. In addition, NLRP1 inhibition has been shown to ameliorate autophagy dysfunction and amyloid beta (Aβ) disposition in models of Alzheimer’s disease, indicating its broader implications in neuroinflammation and cognitive decline [139]. The AIM2 inflammasome has also emerged as a significant player in neuroinflammatory processes associated with depression. Recent findings suggest that AIM2 is activated by various stressors, including LPS, leading to a chronic neuroinflammatory state that contributes to the pathogenesis of depression [140,141]. AIM2 activation has been linked to the production of IL-1β and other inflammatory mediators, which can perpetuate neuroinflammation and impair neurotransmitter systems [140,142].

The interplay between astrocytes and microglia further complicates this picture; astrocytes can release signaling molecules that modulate microglial activation, thereby influencing the overall inflammatory state of the CNS [143]. The study reported by Kano et al. [123] demonstrated that glutathione S-transferase M1 (GSTM1) in astrocytes promotes pro-inflammatory switching between astrocytes and microglia during brain inflammation. The model used for this research involved systemic administration of LPS to induce brain inflammation in mice. GSTM1 was found to be essential for microglia activation and the production of pro-inflammatory mediators in response to inflammatory stimuli. The study demonstrated that astrocyte-specific knockdown of GSTM1 resulted in a significant reduction in microglial activation during LPS-induced brain inflammation. Furthermore, GSTM1 activates the nuclear factor-κB (NF-κB) pathway in astrocytes, which, in turn, triggers the secretion of proinflammatory mediators such as granulocyte-macrophage colony-stimulating factor (GM-CSF) and C-C-motif chemokine ligand 2 (CCL2). This indicates that GSTM1 primes astrocytes for inflammatory responses by enhancing microglia activation through the release of these mediators, thus emphasizing the importance of glial–glial communication in regulating neuroinflammatory processes and their implications for depression [123]. Activation of microglia and astrocytes is not just a response to injury, but can also be influenced by chronic stress. Chronic stress has been shown to induce a proinflammatory state in both microglia and astrocytes, leading to increased production of inflammatory cytokines and exacerbation of depressive symptoms. Leng et al. reported that menin deficiency in astrocytes leads to depressive-like behaviors in mice, suggesting that astrocyte activation is a key factor in neuroinflammation and mood [144]. This indicates that chronic activation of glial cells in response to stress may contribute to the development and persistence of depression.

Transcriptional changes in glial cells are also key to understanding their role in depression. The research has demonstrated that microglial activation is associated with changes in the expression of various homeostatic markers, such as P2Y12, particularly in patients with MDD [145]. These markers are indicators of microglial activation and may influence the inflammatory environment in the brain. For example, the P2Y12 receptor is involved in microglial signaling and has been shown to regulate the inflammatory response, which may further exacerbate depressive symptoms [146]. In addition, the expression of proinflammatory cytokines by activated microglia may lead to changes in neurotransmitter receptor expression, further contributing to the pathophysiology of depression [147]. In addition to microglia, astrocytes also show transcriptional changes in response to inflammatory stimuli. For example, expression of brain-derived neurotrophic factor (BDNF) is often reduced in the context of depression, and this reduction has been linked to altered astrocyte function [148]. BDNF is important for neuronal survival and synaptic plasticity, and its dysregulation may contribute to the development of depressive symptoms. Reduced BDNF levels have been associated with depressive symptoms, and inflammatory cytokines released by activated glial cells inhibit BDNF signaling. This suggests that neuroinflammatory processes mediated by microglia and astrocytes contribute to the neurotrophic deficits observed in depression [131,149]. In addition, astrocytes can influence neurotransmitter receptor expression by releasing gliotransmitters, which can modulate synaptic activity and plasticity [107].

The role of astrocytes in modulating neuronal function is also crucial in understanding the mechanisms of depression. Astrocytes may influence synaptic transmission by regulating the availability of neurotransmitters such as glutamate and modulating synaptic plasticity [123]. Dysregulation of astrocyte function can lead to impaired neurotransmission and contribute to the cognitive deficits often seen in people with depression. For example, Cui et al. identified that the involvement of Kir4.1 in depression lies in its upregulation in astrocytes in the lateral habenula (LHb) of animal models of depression, which contributes to enhanced neuronal hyperpolarization and increased bursting activity, which is associated with depressive-like behaviors. The mechanisms involving Kir4.1 and its effects on neuronal activity are physiologically linked. The mechanism involves Kir4.1 that facilitates the clearance of extracellular potassium (K+), leading to a decrease in extracellular K+ concentration, which directly influences the resting membrane potassium (RMP) of neurons, leading to hyperpolarization of neurons, and activates calcium channels that promote bursting, ultimately affecting monoaminergic centers involved in the pathophysiology of depression [150].

### 3.4. Macrophages and Other Peripheral Immune Cells in CNS Inflammation

In recent years, the understanding of neuroinflammation has expanded significantly, in particular regarding the role of infiltrating immune cells such as monocytes, macrophages, neutrophils, and dendritic cells in inflammatory processes in the brain. The role of monocytes and macrophages in the context of neuroinflammation and depression is increasingly recognized as a critical area of research. These immune cells, which originate in the bone marrow and circulate in the bloodstream, can infiltrate the CNS and contribute to inflammatory processes that impact brain function and mood regulation.

Monocytes are a type of white blood cell that can differentiate into macrophages upon entering tissues, including the brain. In response to neuroinflammatory stimuli, such as infection or injury, monocytes migrate to the site of inflammation, where they can be activated. Recent studies have shown that peripheral T cells infiltrate the brain and influence microglial maturation and phenotype, suggesting a complex interplay between systemic and central immune responses. This infiltrate is particularly relevant in the context of aging, where dysregulated meningeal immunity may exacerbate neuroinflammation and contribute to cognitive decline [151]. Monocyte-derived macrophages have been implicated in the regulation of neuroinflammatory processes, particularly in response to injury or infection [152]. In MDD patients, changes in monocyte subset levels were observed. Studies have shown that patients with MDD often have an increase in proinflammatory monocyte subsets, which are characterized by a higher expression of markers such as CD14+ and CD16+ [128]. These subsets are characterized by their distinct roles in inflammation and immune response. For example, CD14+ monocytes are usually associated with proinflammatory responses, whereas CD16+ monocytes are often linked to tissue repair and anti-inflammatory activities. Studies have shown that people with MDD have an increased frequency of CD14+ monocytes, which correlates with increased levels of proinflammatory cytokines such as IL-6 and tumor-α TNF-α [134,153]. This dysregulation of cytokines is critical because it not only reflects the inflammatory state of the individual but also suggests a potential mechanism by which inflammation may contribute to depressive symptoms. In addition, depressed patients who have never been treated may have defective inflammatory pathways in their monocytes, which may be linked to poor responses to treatment. The chemokine CCL2 (MCP-1) has been observed to be involved in monocyte recruitment and is usually elevated in MDD, reflecting increased monocyte activity and inflammation [154].

Macrophages also play a significant role in neuroinflammation and depression. In the context of depression, macrophages are often activated, leading to the release of proinflammatory cytokines that can disrupt neurotransmitter signaling and contribute to mood dysregulation [155,156]. Furthermore, macrophages can influence the function of other immune cells, including lymphocytes and neutrophils, thereby shaping the overall inflammatory environment in the brain. The activation of macrophages and the subsequent release of proinflammatory cytokines can lead to a cascade of neuroinflammatory processes affecting neurotransmitter systems. For example, elevated levels of IL-6 and TNF-α can disrupt serotonin metabolism by promoting activation of the kynurenine pathway, which diverts tryptophan away from serotonin synthesis and towards the production of neuroactive metabolites (quinolinic acid (QUIN)) [157,158]. QUIN is known to be neurotoxic and may exacerbate neuronal dysfunction, thus contributing to the pathophysiology of depression [159]. In addition, chronic macrophage activation can lead to sustained neuroinflammation, which has been associated with alterations in dopamine and glutamate signaling, further complicating the neurochemical landscape in MDD [160].

The spatial reorganization of macrophages in response to neuroinflammatory stimuli is also notable. Resident microglia, a type of macrophage in the brain, often signalize increased migration and clustering in regions such as the hippocampus, which is critically linked to emotional regulation [161,162]. This spatial reorganization is accompanied by changes in cell signaling pathways that contribute to the pro-inflammatory state characteristic of depression [163]. For example, studies analyzing transcriptomic differences in the hippocampus of depressed individuals have reported upregulation of genes related to the immune response, including cytokines [164,165]. In addition to activation and reorganization of macrophages, transcriptomic profiling reveals significant changes in gene expression patterns that underscore these immune processes. Elevated cytokine gene expression levels in peripheral blood cell samples from MDD patients suggest a systemic inflammatory response that correlates with depressive symptoms [164,165,166]. Studies indicate that peripheral blood monocytes, which can differentiate into macrophages, show altered phenotypes in patients with depression, contributing to the inflammatory milieu [165,166,167]. Such transcriptomic changes reflect the potential impact of chronic stress and consequent depression-related alterations in the immune system [163]. In addition, there is evidence pointing to a complex interplay between macrophage activation, neuroinflammation, and the HPA axis, which is known to be dysregulated in depression. This dysregulation not only increases the production of inflammatory markers but also influences the behavioral manifestations of the disease and depressive symptoms [165,166]. The concept that macrophage-derived cytokines contribute to the neurobiological mechanisms underlying depressive disorders has been termed the “macrophage hypothesis of de-depression”, linking immune activation to the etiology of depression [161,165].

Infiltration of monocytes and macrophages in the CNS is particularly relevant in the context of depression-associated neuroinflammation. The research has shown that chronic stress and systemic inflammation can lead to the recruitment of monocytes to the brain, where they can exacerbate neuroinflammatory processes [168]. Shechter et al. demonstrated that monocyte-derived macrophages play a crucial role in mediating the inflammatory response following spinal cord injury, highlighting their potential impact on neuroinflammation in the brain [168]. This suggests that monocyte infiltration may contribute to the neuroinflammatory milieu underlying depression. Activation of monocytes and macrophages in the brain leads to the release of proinflammatory cytokines, which can further activate microglia and astrocytes, thereby perpetuating neuroinflammation. This interplay between peripheral immune cells and resident immune cells in the brain is key to understanding the mechanisms of neuroinflammation in depression. For example, Yan et al. found that peripheral monocyte-derived cells can counteract amyloid plaque pathogenesis in a mouse model of Alzheimer’s disease, indicating that monocytes can have both beneficial and detrimental effects on brain health, depending on the context [169]. This duality highlights the complexity of monocyte and macrophage functions in the CNS.

Figure 3 illustrates the interaction between peripheral inflammation, brain inflammation, and the HPA axis, highlighting their roles in stress-related disorders. This illustration shows how systemic inflammation can exacerbate brain inflammation and contribute to stress-related disorders, including depression.

The role of monocytes and macrophages in neuroinflammation is further complicated by their interactions with other immune cells and signaling pathways. For example, the choroid plexus, a structure in the brain that produces cerebrospinal fluid, may serve as a gateway for monocyte infiltration into the CNS [170]. Research has shown that monocytes can migrate to the choroid plexus and differentiate into macrophages, which can then influence neuroinflammatory responses in the brain [171,172]. This highlights the importance of understanding the pathways by which monocytes and macrophages interact with other immune cells in the context of neuroinflammation and mood disorders. In addition to their role in promoting inflammation, monocytes and macrophages may also contribute to inflammation resolution and tissue repair.

The interaction between immune cells and neurotransmitter systems is further complicated by the effects of stress and other environmental factors. Chronic stress has been shown to increase BBB permeability, allowing infiltration of peripheral immune cells into the CNS [173,174]. This infiltration can exacerbate neuroinflammation and disturb the delicate balance of neurotransmitters such as serotonin and dopamine [175,176]. While inflammation can lead to depressive symptoms, the presence of depression can also exacerbate inflammatory responses. For example, people with MDD often have elevated levels of inflammatory markers, suggesting that depression itself may contribute to a pro-inflammatory state [155,177]. This cyclical relationship underlines the need for specific interventions to address both neuroinflammatory processes and the underlying mood disorder.

Lymphocytes, particularly T cells, have been implicated in neuroinflammatory processes associated with depression. Activation of Th17 cells, a subset of CD4+ T cells, has been shown to promote depression-like behaviors in animal models, suggesting a direct link between T cell activation and mood disorders [178]. Th17 cells produce proinflammatory cytokines, such as IL-17, which can influence neurotransmitter systems by altering serotonin metabolism and promoting neuroinflammation [4]. In contrast, regulatory T cells (Tregs) have been identified as having a protective role in mood regulation. Depletion of Tregs in mice has been associated with increased anxiety-like behaviors and elevated levels of inflammatory cytokines, suggesting that Tregs may help attenuate the effects of neuroinflammation on mood [173,179]. This dual role of lymphocytes underlines the complexity of their involvement in depression, where certain subsets may exacerbate symptoms while others provide protective effects.

Natural killer (NK) cells, a subset of lymphocytes, have also attracted attention for their potential role in depression. These cells are primarily known for their ability to target and destroy infected viral or tumor cells, but they also produce cytokines that can modulate the immune response. Studies have shown that stress can lead to changes in NK cell activity, which may contribute to depressive symptoms [157,173]. The relationship between NK cells and depression is complex, as their activation can either promote inflammation or exert protective effects, depending on the context. For example, increased NK cell activity has been associated with higher levels of pro-inflammatory cytokines, which may exacerbate depressive symptoms, while their regulatory functions may help restore balance in the immune response [4].

Neutrophils, another key component of the immune response, have also been linked to depression through their role in systemic inflammation. The neutrophil-to-lymphocyte ratio (NLR) has been proposed as a biomarker for depression, with higher ratios indicating increased inflammation [176,180]. Neutrophils release pro-inflammatory cytokines, such as IL-1 and TNF-α, which can disrupt neurotransmitter systems and contribute to the development of depressive symptoms [155]. Neutrophils can act as early reactors in the inflammatory process, triggering the subsequent recruitment of other immune cells, including monocytes and macrophages, thus amplifying the inflammatory response [181]. The temporal dynamics of neutrophil infiltration suggest that neutrophils may play a dual role in both the initiation and resolution of inflammation, depending on the context of the inflammatory stimulus [83].

Dendritic cells, as antigen-presenting cells, are also involved in modulating neuroinflammation. They can migrate to the lesion site and influence the local immune response by presenting antigens to T cells, thereby shaping the adaptive immune response in the CNS [152]. The interaction between dendritic cells and other immune cells, such as macrophages and neutrophils, further emphasizes the complexity of the neuroinflammatory milieu [182]. Furthermore, the recent research has highlighted the importance of chemokines in orchestrating the re-recruitment and activation of these immune cells. The chemokine CCL17 is upregulated under various inflammatory conditions and plays a significant role in mediating monocyte and macrophage infiltration into the CNS. This chemokine signaling is essential for maintaining the balance between proinflammatory and anti-inflammatory responses, which is crucial for preventing chronic neuroinflammation that can lead to neurodegeneration [183].

### 3.5. The Role of the BBB in Mood Disorders

BBB dysfunction is increasingly recognized as a significant factor in the pathophysiology of neuroinflammation and its association with depression. The BBB is a highly selective barrier made up of endothelial cells that line the blood vessels of the brain, together with astrocytic and pericyte legs. The cerebrospinal blood–cerebrospinal fluid barrier (BCSFB), formed mainly by the choroid plexus, plays a key role in maintaining homeostasis in the CNS. This barrier regulates the exchange of substances between the blood and cerebrospinal fluid (CSF), essential for brain health. Neuroinflammation can significantly disrupt the function of the BCSFB and the glia, leading to detrimental effects on neurotransmitter systems and overall brain function. The choroid plexus is a specialized structure located in the ventricles of the brain responsible for the production of CSF. It consists of a layer of epithelial cells tightly bound together by tight junctions, forming the BCSFB. Recent studies have shown that neuroinflammation can lead to morphological and functional changes in the choroid plexus, which, in turn, affect CSF dynamics [184]. The study by Jeong et al. demonstrated that changes in choroid plexus volume were associated with cognitive dysfunction in Parkinson’s disease, suggesting that inflammation-induced changes in the choroid plexus may affect cognitive function by disrupting CSF flow and composition [184]. In addition, the study indicated that increased choroid plexus permeability during neuroinflammatory states facilitates the entry of potentially harmful substances into the CNS, exacerbating neuroinflammation and contributing to depressive symptoms. The glymphatic system, which facilitates the removal of waste products from the brain, is closely linked to BCSFB function. This system relies on the movement of CSF through the interstitial spaces of the brain, mainly mediated by aquaporin-4 (AQP4) channels localized on astrocytic terminal astrocyte legs. Neuroinflammation has been shown to affect glymph function by disrupting AQP4 polarity and reducing its expression, as shown in research by Yu et al. who found that tPA deficiency in mice resulted in decreased AQP4 expression and impaired glymph clearance, leading to increased accumulation of neurotoxic substances such as amyloid-beta (Aβ) [185]. This accumulation is particularly relevant in the context of depression, as elevated Aβ levels have been linked to depressive symptoms and cognitive decline [186]. The interaction between neuroinflammation and the glia is further complicated by the role of microglia. A study by Yang et al. demonstrated that activated microglia can synchronize with the circadian rhythm of the glia, thereby modulating its function [187]. When microglia were depleted, glia clearance improved, indicating that microglial activation during neuroinflammatory states might reduce the efficacy of the glymphatic system. This deficiency leads to the accumulation of neurotoxic metabolites, contributing to the pathophysiology of depression [187]. The implications of glymph dysfunction extend beyond the elimination of waste products; they also include the regulation of neurotransmitter systems. For example, disruptions in glymphatic function lead to altered levels of neurotransmitters such as serotonin and dopamine. A study by Chi et al. showed that *Porphyromonas-gingivalis*-induced neuroinflammation led to cognitive impairment associated with glymphatic dysfunction, suggesting that the interplay between neuroinflammation and glymphatic elimination is essential for maintaining neurotransmitter homeostasis [188].

BBB dysfunction leads to increased permeability, allowing inflammatory mediators and immune cells to infiltrate the brain, which contributes to neuroinflammation and the development of depressive symptoms [189]. Devinney et al. emphasized that a permeable BBB facilitates the infiltration of proinflammatory cytokines and immune cells, which have been shown to lead to cognitive dysfunction and mood disturbances in both animal models and human disorders, including major depression [190]. The presence of these inflammatory mediators in the brain activates microglia and astrocytes, leading to a cascade of neuroinflammatory responses that exacerbate depressive symptoms [189]. The mechanisms underlying BBB dysfunction in the context of neuroinflammation are complex. Proinflammatory cytokines can disrupt the integrity of the tight junctions between endothelial cells, leading to increased BBB permeability. TNF-α has been shown to induce the expression of matrix metalloproteinases (MMPs), which degrade extracellular matrix components and tight junction proteins, thereby compromising BBB integrity. This disruption allows inflammatory mediators and immune cells to pass into the brain, perpetuating the neuroinflammatory cycle associated with depression [191]. In addition to cytokine-mediated effects, oxidative stress is another critical factor contributing to BBB dysfunction. ROS generated during neuroinflammatory processes can damage endothelial cells and disrupt tight junctions, increasing BBB permeability [192]. This oxidative stress can be exacerbated by conditions such as chronic stress, which is known to increase cortisol levels and promote inflammation [189].

The consequences of BBB dysfunction extend beyond the entry of inflammatory mediators; they also impact neuronal function and survival. Increased BBB permeability leads to the accumulation of neurotoxic substances in the brain, which contribute to neuronal damage and loss [189]. The presence of fibrinogen, a protein that does not usually cross the BBB, can activate microglia and promote neuroinflammation, leading to cognitive decline and mood disturbances [193]. This suggests that BBB dysfunction not only facilitates the entry of inflammatory mediators but also contributes to the neurotoxic environment that underlies depression. In addition, BBB dysfunction has been implicated in aging and neurodegenerative diseases, both of which are associated with an increased risk of depression. Therapeutically, the BBB dysfunction approach offers new strategies for treating depression. Agents that improve BBB integrity or inhibit the inflammatory processes that lead to BBB breakdown may attenuate neuroinflammatory responses associated with depression. Compounds such as melatonin have shown promise in maintaining BBB integrity and reducing neuroinflammation, suggesting that they may have therapeutic potential in the context of depression [191].

### 3.6. Neurogenesis and Neuroplasticity in Depression

Neurogenesis, the process of generating new neurons from neural stem cells, occurs mainly in two regions of the adult brain: the hippocampus and the subventricular-lateral area [194]. The hippocampus is particularly important for learning, memory, and emotional regulation, making it a focal point in the study of depression. The topic of adult neurogenesis in the human brain is a complex and nuanced area of research that has generated considerable debate among scientists. While early studies suggested that neurogenesis, particularly in the hippocampus, continues into adulthood, more recent findings indicate that this process may be significantly diminished or even absent in adult humans [195,196,197,198,199]. While neurogenesis is an essential component of brain plasticity in rodents, its presence in adult humans is less clear. The challenges of translating findings from rodent models to humans are compounded by species differences in neurodevelopmental timescales and patterns of neurogenesis. For example, Zeiss [195] points out that while rodents show more pronounced neurogenesis throughout life, human neurogenesis may be limited and context-dependent. Spalding and co-workers [196] showed considerable inter-individual variation in hippocampal neurogenesis among adults, suggesting that while some individuals may exhibit neurogenesis, others may not, complicating the generalizability of findings from animal models to human physiology. In addition, Braun and Jessberger [197] noted that the evidence for adult neurogenesis in humans is still debated, with research continuing to detect markers indicative of neurogenesis in the adult human hippocampus. Historically, the discovery of adult neurogenesis in different species, including rodents and non-human primates, led to the hypothesis that similar processes might occur in humans [198]. An initial study, such as that by Eriksson et al. [199], provided evidence for the presence of new neurons in the adult human hippocampus, suggesting that neurogenesis plays a role in memory and learning. In contrast, the study by Sorrells et al. presented data indicating that neurogenesis in the adult human hippocampus is rare or absent, with markers typically associated with newborn neurons not being uniquely expressed in the dentate gyrus [200]. This aligns with the conclusions drawn by Dennis et al. who observed that neurogenesis in adults is functionally insignificant, declines rapidly with age, and may reach negligible levels by middle age [201].

The debate on adult neurogenesis in humans is further complicated by methodological differences in the way studies assess neurogenesis. For example, post-mortem analyses have shown different results, with some studies indicating neurogenesis continues into old age, while others report a marked decline or absence of new neurons in adults [202,203]. Discrepancies may arise from the techniques used to identify neurogenic markers, as emphasized by Terstege et al. who cautioned against relying solely on post-mortem tissue analysis without considering the dynamic nature of neurogenesis [202]. In addition, the role of environmental factors such as stress and enriched environments has been shown to influence neurogenesis, suggesting that the context in which neurogenesis is studied may produce different outcomes [204].

Another level of complexity is added by the potential functional implications of neurogenesis. While some researchers argue that even low levels of neurogenesis may contribute to cognitive functions such as memory and mood regulation, others argue that the absence of significant neurogenesis in adults may limit the brain’s capacity for self-repair and adaptation [205,206]. For example, studies have linked reduced neurogenesis to cognitive decline and neurodegenerative diseases, suggesting that understanding the mechanisms behind neurogenesis could have therapeutic implications [207,208]. In addition, the neuroplasticity hypothesis of depression implies that deficits in neuroplasticity, rather than the complete absence of neurogenesis, may be more relevant to understanding the disorder. Price and Duman [209] argue that neuroplasticity is fundamental to adaptive function. They emphasize that deficits in neuroplasticity are often observed in depressive disorders, suggesting that the mechanisms underlying these deficits may be more complex than a simple lack of neurogenesis. Supporting this view, Noda et al. [210] demonstrated that depression is associated with reduced structural plasticity in key brain regions, such as the prefrontal cortex and hippocampus, which are crucial for mood regulation and cognitive function.

In the context of depression, deficiencies in the processes of neurogenesis and neuroplasticity are increasingly recognized as a significant contributor to the pathophysiology of this disorder. Research has shown that stress and depression are associated with reduced neurogenesis in the hippocampus, which contributes to the cognitive deficits and mood disturbances characteristic of this disorder. Zhou et al. demonstrated that chronic stress exposure leads to a significant decrease in hippocampal neurogenesis, which is correlated with the development of depressive-like behaviors in animal models [211]. The relationship between neurogenesis and depression is complex. While some studies suggest that impaired neurogenesis directly contributes to depressive symptoms, others suggest that neuroplasticity, the brain’s ability to reorganize and form new synaptic connections, may be equally, if not more important in the context of depression [212]. Neuroplasticity encompasses diverse processes, including synaptogenesis, dendritic remodeling, and changes in synaptic strength, all of which are essential for learning and memory. Impaired neuroplasticity leads to reduced cognitive flexibility and emotional resilience, further exacerbating depressive symptoms [213]. The role of BDNF in promoting neurogenesis in the hippocampus is crucial [214]. Low BDNF levels have been associated with depression, and studies have shown that antidepressant treatments can increase BDNF levels, promoting neurogenesis and improving mood [215]. Furthermore, the effects of oxidative stress on neurogenesis and neuroplasticity cannot be overlooked. High levels of ROS damage neuronal cells and impair processes of neurogenesis and synaptic plasticity [216]. Oxidative stress has been shown to inhibit the proliferation of neural progenitor cells and disrupt signaling pathways involved in neuroplasticity, further contributing to the cognitive and emotional deficits observed in depression [217]. This emphasizes the importance of targeting oxidative stress as a potential therapeutic approach for restoring neurogenic and neuroplastic functions in depressed patients.

### 3.7. Oxidative Stress and Its Impact on Neurotransmitter Systems

Oxidative stress and neurotransmitter alterations are key components of the biological mechanisms underlying neuroinflammation in depression. Oxidative stress refers to an imbalance between the production of ROS and the body’s ability to detoxify these harmful compounds or repair the resulting damage. This imbalance leads to cell damage and inflammation, which are closely linked to the pathophysiology of depression. The brain is particularly vulnerable to oxidative stress due to its high metabolic activity and oxygen consumption. High levels of ROS damage cellular components including lipids, proteins, and DNA, leading to neuronal dysfunction and death. In the context of depression, oxidative stress has been shown to contribute to the neuroinflammatory processes that exacerbate depression. Liu et al. demonstrated that chronic mild stress-induced oxidative stress is associated with increased levels of proinflammatory cytokines and alterations in neurotransmitter systems, suggesting a direct link between oxidative stress and neuroinflammation in depression [218].

Neurotransmitter alterations are a hallmark of depression, with significant changes seen in key neurotransmitter systems including serotonin, dopamine, and norepinephrine. These neurotransmitters play essential roles in mood regulation, and their dysregulation is implicated in the development of depressive symptoms. High levels of ROS lead to the oxidation of neurotransmitters, affecting their function and availability. This oxidative change can lead to reduced levels of serotonin and dopamine, contributing to the mood dysregulation characteristic of depression. Furthermore, the impact of oxidative stress on neurotransmitter systems extends beyond direct oxidative damage. For example, oxidative stress can influence the expression and function of neurotransmitter receptors, further disrupting synaptic signaling. Research has shown that oxidative stress leads to downregulation of serotonin receptors, affecting serotonergic signaling and contributing to mood disorders. In addition, oxidative stress has been linked to alterations in the glutamatergic system, which is essential for synaptic plasticity and cognitive function. The dysregulation of glutamate signaling has been implicated in the development of depression, underlining the importance of understanding the interplay between oxidative stress and neurotransmitter systems [219].

The role of antioxidants in mitigating oxidative stress and its effects on neurotransmitter systems has attracted attention in the context of depression. Antioxidants can neutralize ROS and protect against oxidative damage, potentially alleviating depressive symptoms. Supplementation with antioxidants, such as N-acetylcysteine (NAC), can improve mood and reduce oxidative stress in people with depression [220]. This suggests that targeting oxidative stress through antioxidant therapies may offer a promising approach to treating depression.

In general, the interaction between oxidative stress and neurotransmitter alterations is a key factor to consider in the pathophysiology of depression. Neuroinflammation, characterized by the release of proinflammatory cytokines and activation of glial cells, exacerbates oxidative stress, leading to significant disruptions in neurotransmitter systems. This dysregulation not only contributes to the symptoms of depression but also affects neurogenesis and synaptic plasticity, further complicating the clinical picture.

## 4. The Gut–Brain Axis and Neuroinflammation in Depression

### 4.1. Gut Microbiota Composition in Depression

Gut microbiota composition has been increasingly recognized as a significant factor in the pathology of depression. It has been argued that changes in the gut microbiota may influence mood disorders, including MDD. These changes may affect neurobiological and behavioral outcomes, reinforcing the importance of maintaining gut homeostasis for mental health. For example, research by Xu et al. indicates that disturbances in the gut microbiota may lead to metabolic changes relevant to depressive-like behaviors, highlighting in particular glycerophospholipid metabolism along the gut–brain axis as a mediating factor in this phenomenon [221]. Similarly, Kazemian et al. explored how early-life adversity negatively affects the diversity of the gut microbiota, leading to changes that may predispose individuals to mood disorders later in life, thus supporting the hypothesis of a link between gut health and depression [222]. In another study, it was reported that *Coprococcus* and *Faecalibacterium* species are less prevalent in people with depression, a finding confirmed by other research on gut flora, which studied people with depression compared to healthy controls [223]. This aligns with findings from randomized controlled trials showing that probiotics can positively alter gut microbiota and contribute to psychological health [224]. Additionally, high microbiome diversity has been linked to improved emotional well-being, supporting the idea that a healthier gut microbiome correlates with better mental health outcomes [225].

The interactions between the gut microbiome and the brain are elucidated through the gut–brain axis, which serves as a communication channel influencing the functions of the central nervous system. For example, certain gut bacteria produce neuroactive metabolites and help modulate inflammatory pathways, crucial in depressive disorders [226,227]. Studies have also indicated that gut dysbiosis can lead to behavioral changes associated with depression, suggesting that microbial imbalances may directly affect brain function and emotional regulation [228,229,230].

Zhao et al. advocate a multi-omics approach examining the impact of the gut microbiome on major depressive disorder, suggesting an interplay between metabolic and immune alterations due to changes in the microbiota, although the question of causality versus consequence remains an open investigation [231]. Meanwhile, Liang et al. provide a comprehensive review of the potential mechanisms linking dysbiosis to depressive symptoms, recognizing that intestine-microbiome-brain interactions could lead to neurotransmitter dysregulation and alterations in the stress response, both crucial in the pathophysiology of depression [232]. Supporting these findings is research by Qu et al., who have used animal models to illustrate the parallels between social-stress-induced alterations in the gut microbiota and depressive states, specifically noting the importance of certain families of gut bacteria, such as *Ruminococcaceae*, in metabolism [233]. Furthermore, Luo et al. highlighted the link between gut microbiota and the neuroendocrine axis, revealing that specific changes in gut flora, particularly an increase in *Actinobacteria* and a decrease in *Bacteroidetes*, correlate with significant behavioral changes and potential depressive phenotypes by modulating glucocorticoid response pathways [234].

A systematic review by Cheung et al. emphasizes the functional aspects of the microbiome, suggesting that the metabolic outcomes of the microbiome community may be of more importance than mere species composition in understanding its link to depression [235]. Furthermore, Bilenduke et al. point to potential cognitive ramifications arising from alterations in the microbiota resulting from both the direct effects of diseases such as cancer and treatment modalities such as chemotherapy [236]. In addition, McGuinness et al. point out that the role of gut microbiota in the regulation of depressive symptoms extends beyond microbial diversity, indicating that dietary influences significantly affect gut composition and subsequent behavioral outcomes [237]. Environmental factors, including diet, play a significant role in changing the gut microbiota. Dietary habits, together with lifestyle choices, can lead to variations in the composition of the gut microbiome, with implications for mental health. Dietary interventions have been proposed as potential adjunctive treatments for depressive disorders, with some studies suggesting that certain dietary patterns may lead to changes in the microbiota that improve mental health [238,239]. Probiotic treatments are beneficial not only for restoring microbial balance but also as viable options for tackling anxiety and depressive symptoms [240,241]. Furthermore, investigations into microbial therapeutic approaches, such as fecal microbiota transplantation, have gained ground due to studies suggesting that bacterial transfers from depressed individuals to germ-free animals produce similar adverse behavioral outcomes [242,243].

### 4.2. Gut Permeability and Its Effect on Neuroinflammation

Intestinal permeability, often referred to in the context of the “permeable gut”, is a significant contributor to a multitude of neuroinflammatory processes central to various neurodegenerative diseases. The gut–brain axis, characterized by bidirectional communication between the CNS and the gastrointestinal tract, emphasizes the importance of gut microbiota composition in maintaining neuronal health and response to inflammation. The correlation between gut permeability and neuroinflammation arises from the fact that disturbances of the gut microbiota, often superimposed on conditions of dysbiosis, cause systemic and localized inflammatory responses, culminating in neuroinflammatory conditions. Changes in gut permeability can have a significant impact on systemic health, including neurological conditions. Castro et al. emphasize that increased intestinal permeability allows cytokines and bacterial products to pass into the circulation, which may eventually disrupt the BBB and promote neuroinflammation [244]. Wang et al. further corroborate this notion, showing that methamphetamine exposure leads to increased intestinal permeability and further neuroinflammation, illustrating a direct pathway from intestinal dysbiosis to neuroinflammatory outcomes [245].

Assessment of the role of gut microbiota in neuroinflammation relies primarily on the involvement of specific bacterial species, particularly in cases involving gut dysbiosis that exacerbates the release of neuroinflammatory cytokines. For example, research by Song et al. highlights that the administration of specific probiotic strains, such as *Roseburia hominis*, can alleviate neuroinflammation by producing short-chain fatty acids (SCFAs), such as propionate and butyrate, which play roles in inhibiting histone deacetylase, thus providing protective effects in neuroinflammatory diseases [246]. This highlights the potential of dietary modulation of gut flora to exert neuroprotective effects via metabolic by-products.

The interplay between gut microbiota and neuroinflammation has been summarized by representing gut microbiota dysbiosis as a contributing factor to neuroinflammatory conditions such as Alzheimer’s disease [247]. Gut-derived immune cells, including T lymphocytes, appear to have roles in modulating inflammatory states in the brain, suggesting not only a link between alterations in gut microbiota and neuroinflammatory responses but also a potential mediating role for the immune system itself [248]. Wider ramifications of gut dysbiosis have been reported, particularly concerning conditions such as intracerebral hemorrhage, indicating that gut microbiota dysbiosis may significantly influence neuroinflammation, as evidenced by increased cytokine levels and subsequent deterioration of neurological functions post-injury [248]. This aligns with the findings of Liu et al. who demonstrated that alterations in gut microbial interactions can lead to significant inflammatory responses in the context of neuroinflammatory disorders [249]. Moreover, the interplay between the microbiome, immune responses, and neuroinflammation reveals a complex dynamic. The research by Wang et al. provides considerable evidence that remodeling the gut microbiota can reduce Alzheimer’s-disease-associated neuroinflammation through alterations in immune cell functionality, specifically showing how gut-derived inflammatory cytokines interact with CNS immunological responses [250]. Similarly, Wu et al. have illustrated how dietary interventions using selenium peptides can alleviate neuroinflammation, suggesting that gut health communicates positively with CNS health [251]. These findings collectively support the hypothesis that interventions targeting gut permeability and microbial composition may produce beneficial outcomes in alleviating neuroinflammation-related disorders.

The mechanism of the impact of gut microbiota on neuroinflammation has been linked to various immune pathways, focusing in particular on the role of TLRs and inflammasome activation. For example, activation of the NLRP3 inflammasome has been shown to play a key role in stimulating neuroinflammation in models of Alzheimer’s disease, pointing to a microbiota–gut–inflammasome–brain axis that is crucial for understanding these cross-communication mechanisms [252,253]. In support of this view, studies have shown that increased intestinal permeability following brain injury leads to increased systemic inflammation and neuroinflammatory responses, in particular via the TLR4 pathway [254]. This pathway appears to be a significant mediator for the translocation of gut-derived endotoxins, further affecting neuroinflammatory states.

Modulation of the immune system by the gut microbiota cannot be ignored. Gut dysbiosis not only influences local gut health but can invoke systemic inflammatory responses through the secretion of proinflammatory cytokines, significantly affecting communication and CNS health [255]. Such systemic inflammation can cause neuronal damage, illustrating the complexity of the gut–brain relationship and reiterating the importance of maintaining gut integrity to prevent neuroinflammatory disorders. Strategies aimed at maintaining or restoring gut microbial balance, such as probiotics, prebiotics, and dietary modifications, are both reasonable approaches and potentially essential components of a therapeutic regimen against neuroinflammatory diseases.

### 4.3. Microbial-Derived Metabolites and Their Role in Neurotransmitter Production

Microbial-derived metabolites play a crucial role in the production and regulation of neurotransmitters, significantly influencing brain function and mental health. The gut microbiota comprises trillions of microorganisms that contribute to the synthesis of neurotransmitters, including serotonin (5-HT), dopamine, gamma-aminobutyric acid (GABA), and norepinephrine (NE) [256,257,258]. Specific bacterial species, such as *Lactobacillus* and *Bifidobacterium*, are well documented for their ability to produce these neurotransmitters, suggesting a link between gut health and neurochemical balance in the CNS [256,258]. Microbial metabolites such as SCFAs, including acetate, propionate, and butyrate, are produced by gut bacteria from dietary fiber, essential for maintaining communication between the gut and the brain [259,260,261]. It has been shown to affect neurotransmitter production at both peripheral and central levels, modulating neurophysiological functions and potentially impacting mood and anxiety-related behaviors [260,261,262]. For example, butyrate has been implicated in maintaining the integrity of the BBB and increasing neurogenesis through different signaling pathways affecting neurotransmitter synthase [260,263,264]. In addition, these metabolites may influence systemic inflammation, which has effects on neurochemical pathways involved in mood regulation and cognitive functions [258,265].

Another critical metabolite is derived from the amino acid tryptophan via gut-bacteria-mediated pathways that facilitate its conversion to serotonin, a neurotransmitter widely recognized for its role in mood regulation. Huang and Wu discuss how certain bacterial strains can increase the availability of tryptophan, subsequently enhancing serotonin signaling in the CNS [266]. Modulation of serotonin levels by gut microbial activity may directly affect behaviors associated with mood disorders, illustrating the potential of targeted therapeutic interventions in conditions such as anxiety and depression [267,268].

Indole, a pivotal microbial metabolite, shows promise for its neuromodulatory functions. Indole may influence the emotional behavior of animal models, effectively linking the emotional regulatory effects of intestinal metabolites to microbiota profiles [269]. In addition, the influence of the maternal microbiome on fetal neurodevelopment emphasizes the importance of microbial metabolites in shaping neurotransmitter systems even before birth [270]. Metabolites such as indole-3-propionic acid (IPA) have emerged as critical modulators of neuronal circuit development, demonstrating how microbiome-derived substances shape brain maturation in early life [271].

Microbial metabolites can engage with different immune pathways, thereby influencing neuronal inflammation, which is relevant in many neurodegenerative diseases, including Parkinson’s disease. The research by Sampson et al. highlights how inflammatory responses mediated by gut microbiota may affect dopaminergic signaling, linking gut health to motor control and overall neuronal health through neurotransmitter dynamics [260,263]. In addition, microbial interactions via G-protein-coupled receptors (GPCRs) on host cells provide a molecular framework through which these metabolites exert their effects. Colosimo et al. elucidate how microbial metabolites may act as ligands for these receptors, thereby modulating neurotransmitter signaling and influencing potential behavioral outcomes via the gut–brain axis [272]. Bidirectional GPCR signaling strengthens the connections between gut microbial activity and the central nervous system, making it a focal point for further exploration in therapeutic contexts [273].

Gut-derived neurotransmitters can interact with the enteric nervous system and vagus nerve, indirectly influencing brain activity even if they do not cross the blood–brain barrier [274,275]. For example, the regulation of 5-HT levels in the gut is significantly improved by the presence of a specific gut microbiota, which has an impact on general mental health and cognition [276]. In addition, disturbances in the composition of the gut microbiota (dysbiosis) can lead to altered neurotransmitter metabolism, contributing to the development of neuropsychiatric disorders such as depression and anxiety [258,262,265]. When gut microbial composition is disturbed, changes in neurotransmitter levels and associated mood disturbances occur. These imbalances can lead to decreased levels of key neurotransmitters such as GABA and serotonin, further exacerbating mental health problems [264,265]. Associated behavioral influences from dysbiosis highlight the critical importance of maintaining microbial balance for psychological well-being.

## 5. Clinical Implications: From Mechanisms to Treatments

### 5.1. The Role of Anti-Inflammatory Drugs in Depression

The therapeutic potential of anti-inflammatory drugs in depression, with randomized clinical trials (RCTs) (Table 1) evaluating their efficacy in reducing depressive symptoms, was also explored. This section provides a summary of the most recent RCTs investigating anti-inflammatory agents for depression, including their mechanisms of action, clinical outcomes, and implications for treatment.

Minocycline, a tetracycline antibiotic with anti-inflammatory properties, was evaluated in a randomized clinical trial as an adjunctive treatment for patients with treatment-resistant depression (TRD). The study found that six weeks of minocycline (200 mg daily) significantly improved depressive symptoms compared to placebo, suggesting its potential as a therapeutic option for patients with inflammatory-related depression [277].

A randomized, double-blind, placebo-controlled study assessed the effects of ashwagandha root extract on stress and anxiety in healthy adults. Participants receiving ashwagandha (500 mg daily) exhibited significant reductions in stress levels, increased serotonin levels, and decreased cortisol compared to the placebo group. These findings highlight the adaptogenic and anti-inflammatory effects of ashwagandha, which may be beneficial in stress-induced depressive disorders [278].

Curcumin, a polyphenol with anti-inflammatory and antioxidant properties, was tested in a randomized controlled trial involving obese patients with type 2 diabetes and comorbid depression. After 12 months, patients receiving curcumin showed significant improvements in depression severity, alongside reduced levels of pro-inflammatory cytokines such as IL-1β, IL-6, and TNF-α. This suggests that curcumin’s effects on both metabolic and inflammatory pathways may underlie its antidepressant potential [279].

Empagliflozin, a sodium-glucose co-transporter-2 (SGLT2) inhibitor commonly used for diabetes management, was tested as adjunctive therapy to citalopram in a randomized trial. Over eight weeks, patients receiving empagliflozin exhibited greater reductions in depression severity compared to the placebo group, supporting the hypothesis that metabolic and inflammatory modulation may enhance antidepressant response [280].

The Youth Depression Alleviation with Anti-inflammatory Agents (YoDA-A) trial investigated aspirin (100 mg daily) and rosuvastatin (10 mg daily) as adjunctive treatments in young people with moderate to severe MDD. The trial found no significant difference in depressive symptom reduction between aspirin, rosuvastatin, and placebo. However, exploratory analyses suggested rosuvastatin may have some secondary benefits on quality of life and global impression scales, warranting further investigation [281].

A large-scale randomized clinical trial investigated whether low-dose aspirin (100 mg daily) could prevent depression in older adults. The results showed that aspirin did not significantly reduce the risk of developing depression compared to placebo, indicating that anti-inflammatory interventions may not be effective for primary prevention in this population [282].

A randomized controlled trial explored the effects of adding celecoxib, a selective cyclooxygenase-2 (COX-2) inhibitor, to vortioxetine in patients with MDD and elevated inflammatory markers. The study found that patients receiving the combination therapy exhibited greater reductions in depressive symptoms compared to those receiving vortioxetine alone. This supports the potential role of targeted anti-inflammatory augmentation in subtypes of depression associated with systemic inflammation [283].

A hybrid parallel-arm randomized trial evaluated the anti-inflammatory effects of low-dose naltrexone in MDD patients stratified by inflammatory status. Patients receiving naltrexone demonstrated significant improvements in depressive symptoms, particularly those with elevated CRP levels, suggesting that inflammation-modulating treatments may be particularly beneficial in inflammatory subtypes of depression [284].

NAC, an antioxidant with anti-inflammatory properties, was investigated as an adjunct to standard antidepressant treatment in patients with TRD and elevated inflammatory markers. The study found that NAC supplementation led to significant improvements in depressive symptoms and reductions in oxidative stress markers, supporting its potential role in inflammation-driven depression [285].

A randomized placebo-controlled trial evaluated glycyrrhizic acid (GZA), a bioactive compound with anti-inflammatory effects, as an adjunct to selective serotonin reuptake inhibitors (SSRIs). Patients receiving GZA demonstrated significantly greater improvements in depression severity, with higher response rates among those with baseline elevated inflammation, reinforcing the potential for anti-inflammatory augmentation strategies in specific depression subtypes [286].

Overall, the reviewed studies highlight the potential of various anti-inflammatory agents in the treatment of depression, particularly in patients with elevated inflammation. While some interventions, such as minocycline, curcumin, and NAC, have demonstrated significant antidepressant effects, others, including aspirin, have yielded negative results. These findings underscore the importance of identifying inflammation-driven subtypes of depression to tailor treatment approaches effectively.

### 5.2. Current and Emerging Biomarkers for Personalized Treatment Approaches

While classical inflammatory markers such as IL-6, TNF-α, and CRP have been widely studied, recent research has focused on identifying novel biomarkers that may offer greater specificity and predictive power in personalized treatment approaches for depression. Figure 4 depicts novel inflammatory biomarkers for depression.

#### 5.2.1. Interleukin-1 Family: IL-1β and IL-1RA

The IL-1 family has gained attention for its role in neuroinflammation and depression. Elevated levels of IL-1β have been linked to increased depression severity and treatment resistance, making it a potential biomarker for identifying patients who may benefit from anti-inflammatory therapies [287]. In contrast, IL-1 receptor antagonist (IL-1RA) has been found to have neuroprotective effects by counteracting the pro-inflammatory actions of IL-1β. Higher IL-1RA levels have been correlated with improved treatment responses, suggesting its role as a potential biomarker for antidepressant efficacy [288].

IL-17, particularly IL-17A, has gained attention for its role in the immune response and its potential link to depression. The research indicates that IL-17A may induce depression-like behaviors by activating signaling pathways such as NF-κB and p38 MAPK, which mediate inflammatory responses in the brain [111,289. In a rodent study, mild cumulative stress led to increased levels of IL-17, which was associated with promoting depression-like behaviors [289]. In addition, elevated IL-17 levels have been correlated with depressive symptoms in different populations, including community-dwelling women and older adults [290,291]. This suggests that IL-17 may serve as a biomarker for depression and a potential therapeutic target. Moreover, IL-17 has been shown to enhance the production of other proinflammatory cytokines, thereby contributing to a vicious cycle of neuroinflammation [289,292]. In addition, the presence of comorbid conditions, such as chronic pain or autoimmune disorders, can exacerbate the inflammatory response and further complicate the clinical picture of depression [293,294]. This highlights the need to consider a broader spectrum of inflammatory mediators when investigating the cytokine hypothesis in depression.

#### 5.2.2. Chemokines: CCL2 and CXCL12

Chemokines play a crucial role in immune cell trafficking and have been implicated in depression. C-C motif chemokine ligand 2 (CCL2), also known as monocyte chemoattractant protein-1 (MCP-1), is significantly elevated in individuals with MDD. CCL2 has been associated with heightened neuroinflammation and blood–brain barrier dysfunction, making it a promising target for biomarker research [295]. Similarly, C-X-C motif chemokine ligand 12 (CXCL12) has been found to influence neuroplasticity and synaptic remodeling, and its dysregulation has been linked to depressive symptoms [296].

#### 5.2.3. Interferon-Gamma (IFN-γ)

Interferon-gamma (IFN-γ) is a pro-inflammatory cytokine that modulates the immune response and has been implicated in the development of depression. Studies have shown that elevated IFN-γ levels correlate with anhedonia and cognitive impairment in MDD patients, suggesting its potential use as a biomarker for specific depression subtypes [297]. Moreover, IFN-γ levels have been associated with resistance to conventional antidepressants, highlighting its relevance in precision medicine [298].

#### 5.2.4. Transforming Growth Factor-Beta (TGF-β)

Transforming growth factor-beta (TGF-β) is an anti-inflammatory cytokine that regulates immune homeostasis. Recent studies suggest that individuals with MDD exhibit decreased levels of TGF-β, which may contribute to unchecked neuroinflammation [299]. Low TGF-β levels have also been linked to higher suicide risk, emphasizing its potential as a prognostic biomarker [300].

#### 5.2.5. Soluble Tumor Necrosis Factor Receptors (sTNFRs)

Soluble tumor necrosis factor receptors (sTNFR1 and sTNFR2) have emerged as biomarkers that reflect TNF-α activity more accurately than TNF-α itself. Increased levels of sTNFRs have been observed in patients with treatment-resistant depression and have been proposed as indicators for anti-inflammatory interventions [301]. Their stability in circulation makes them promising candidates for clinical applications.

#### 5.2.6. Neopterin

Neopterin, a marker of immune system activation, has been linked to neuroinflammation in depression. Elevated neopterin levels have been associated with increased oxidative stress and cognitive dysfunction in MDD patients. This biomarker may help identify individuals at risk for chronic-inflammation-related depression [302].

#### 5.2.7. Kynurenine Pathway Metabolites

The kynurenine pathway is a major route of tryptophan metabolism, and its dysregulation has been implicated in depression. Increased levels of kynurenine and quinolinic acid, coupled with reduced kynurenic acid, have been associated with heightened neurotoxicity and inflammation in MDD [303]. This imbalance may serve as a valuable biomarker for treatment stratification.

The identification of novel inflammatory biomarkers offers promising avenues for personalized treatment approaches in depression. Biomarkers such as IL-1β, IFN-γ, CCL2, sTNFRs, and kynurenine metabolites provide deeper insights into the pathophysiological mechanisms of depression and may help guide targeted therapeutic strategies. Future research should focus on integrating these biomarkers into clinical practice to improve diagnostic accuracy and treatment outcomes.

### 5.3. How Depression Therapies Target Neuroinflammation

Cortisol, a glucocorticoid hormone produced by the adrenal glands, plays a key role in regulating various physiological processes, including metabolism, immune response, and the body’s reaction to stress. As the primary glucocorticoid produced by the adrenal glands, it is released in response to stress by activating the HPA axis. Acute stress usually leads to a transient increase in cortisol levels, which can exert anti-inflammatory effects by modulating immune responses. Yeager et al. demonstrated that cortisol can achieve maximal suppression of inflammation at physiological concentrations, suggesting a biphasic relationship in which higher cortisol levels are not necessarily correlated with increased anti-inflammatory effects [304]. Due to its anti-inflammatory properties, cortisol modulates immune responses and maintains homeostasis during stress. Under normal conditions, cortisol helps suppress the production of proinflammatory cytokines and promotes the expression of anti-inflammatory mediators [305]. However, chronic stress can lead to sustained increases in cortisol levels, leading to a state of glucocorticoid resistance, in which the body becomes less responsive to the regulatory effects of cortisol. Ranganathan et al. found that patients with severe traumatic brain injury had significantly elevated cortisol levels, which correlated with increased neuroinflammation and poor long-term outcomes [82]. This suggests that hypercholesterolemia may promote a pro-inflammatory state that may contribute to the pathophysiology of depression [306]. Prolonged exposure to high levels of cortisol can lead to neurotoxicity, particularly in the hippocampus, resulting in impaired neurogenesis and altered neurotransmitter signaling [307]. This neurotoxic effect is associated with a decrease in levels of serotonin (5-HT), the key neurotransmitter involved in mood regulation [308]. The serotonergic system is particularly sensitive to stress, and alterations in serotonin signaling can contribute to depressive symptoms. Furthermore, elevated cortisol levels may lead to increased release of norepinephrine from the locus coeruleus, the primary source of norepinephrine in the brain [309]. This increase in norepinephrine may enhance excitatory neurotransmission, particularly in regions associated with stress responses, such as the rostral linear nucleus [310]. Chronic stress and the resulting hypercortisolism can disrupt norepinephrine balance, leading to symptoms of anxiety and depression [307]. Stress-induced increases in cortisol may also affect dopaminergic signaling, which can contribute to anhedonia [307]. Stress hormones may influence the sensitivity of dopamine receptors, in particular the D2 receptor subtype involved in the pathophysiology of depression [311]. Chronic stress can lead to changes in dopamine receptor expression and function, further exacerbating depressive symptoms.

The impact of cortisol on inflammation is also evident in the context of neurodegenerative diseases. Elevated cortisol levels have been linked to increased neuroinflammation in conditions such as Alzheimer’s disease, in which inflammation is considered a feature of the disease process [312]. Conditions such as Cushing’s syndrome, characterized by abnormally high cortisol levels, exemplify the pathological consequences of excessive cortisol. Patients with Cushing’s syndrome often have elevated inflammatory markers and a higher risk of metabolic syndrome and cardiovascular disease [313]. Conversely, adrenal insufficiency, which leads to low cortisol levels, can also lead to inflammatory dysregulation, as insufficient cortisol cannot adequately suppress inflammatory responses [314]. This duality underscores the need for a nuanced understanding of the role of cortisol in inflammation, particularly in the context of neuroinflammation and its implications for mental health disorders such as depression.

A comprehensive examination of the effects of cortisol in different physiological conditions, including acute and chronic stress, as well as specific disorders such as Cushing’s syndrome and adrenal insufficiency, is essential for elucidating its role in neuroinflammation and depression. The complex relationship between cortisol and inflammation is shown schematically in Figure 5. During acute stress, increased cortisol levels lead to anti-inflammatory effects. However, chronic stress can lead to elevated cortisol levels, which can lead to cortisol resistance and pro-inflammatory responses. Under pathological conditions, Cushing’s syndrome (hypercortisolism) and adrenal insufficiency (hypocortisolism) further emphasize the deregulation of the effect of cortisol on immune functions, leading to exacerbated or uncontrolled inflammation.

The interaction between cortisol, neurotransmitter systems, and neuroinflammation is also noteworthy. Neuroinflammation, characterized by glial cell activation and the release of proinflammatory cytokines, can disrupt neurotransmitter signaling and contribute to the development of depression [315]. Cortisol has been shown to modulate anti-inflammatory responses, and chronic stress can lead to a state of glucocorticoid resistance, in which the normal anti-inflammatory effects of cortisol are diminished [316,317,318]. This resistance can exacerbate neuroinflammation, further affecting neurotransmitter systems and contributing to the cycle of stress and depression.

Therapeutically, targeting cortisol and stress hormone pathways may offer new possibilities for treating depression. For example, interventions aimed at reducing cortisol levels, such as cognitive-behavioral therapy (CBT) and mindfulness-based stress reduction, have shown promise in relieving depressive symptoms [123]. In addition, pharmacological agents that modulate the HPA axis or target inflammatory pathways may provide effective treatment options for people with depression characterized by elevated cortisol and inflammation [134].

The cytokine hypothesis has important implications for treatment strategies for depression. Addressing inflammation using anti-inflammatory agents has shown promise in alleviating depressive symptoms. For example, studies have shown that the use of non-steroidal anti-inflammatory drugs (NSAIDs) and cytokine inhibitors can improve mood and reduce depressive symptoms in people with elevated inflammatory markers [319,320,321,322,323,324]. In addition, lifestyle interventions have been found to reduce neuroinflammation and improve mood by stimulating neuroplasticity and promoting the release of neurotrophic factors in the PFC, a region integral to emotional regulation, decision-making, and cognitive functions and particularly sensitive to inflammatory processes, which can disrupt neurotransmitter systems and contribute to the development of depressive symptoms [79,172,325,326,327,328,329].

Hippocampal neuroinflammation has emerged as a critical factor in understanding the pathophysiology of depression. The hippocampus is particularly vulnerable to inflammatory processes that can disrupt its function and contribute to depressive symptoms. The hippocampus also shows structural changes in response to neuroinflammation, which can be observed in different models of depression. For example, Gold et al. reported that patients with multiple sclerosis and depressive symptoms had smaller hippocampal volumes, suggesting that neuroinflammation may lead to neurodegeneration in this region [330]. This atrophy may be related to chronic activation of microglia and the eventual release of neurotoxic substances that affect neuronal integrity and function [331,332,333,334,335,336,337]. Exercise and other interventions have been shown to mitigate the effects of hippocampal neuroinflammation on depression. Running exercise has been reported to reduce microglial activation and improve neuroinflammatory profiles in hippocampal models of stress-induced depression [338,339,340,341]. This suggests that physical activity may promote neuroprotection and enhance neurogenesis, counteracting the damaging effects of neuroinflammation [338,341]. Similarly, pharmacologic agents that target neuroinflammatory pathways, such as COX-2 inhibitors, have demonstrated similar antidepressant effects by reducing neo-inflammation and restoring normal hippocampal function [342,343,344].

The interaction between the locus coeruleus and other regions of the brain is key to understanding its role in mood regulation. The locus coeruleus receives information from different regions of the brain, including the hypothalamus and amygdala, which are involved in regulating stress and emotional responses [345]. Neuroinflammation can disrupt this communication, leading to altered neuronal connectivity and impaired emotional regulation. Studies have shown that neuroinflammation can lead to increased excitability of locus coeruleus neurons, which may contribute to increased anxiety and depressive symptoms [346]. Furthermore, the role of the locus coeruleus in modulating neuroinflammation is significant. Norepinephrine released by locus coeruleus neurons may influence microglia and astrocyte activity. The research has shown that norepinephrine can have both pro-inflammatory and anti-inflammatory effects, depending on the context and the receptors activated [347]. For example, activation of beta-adrenergic receptors by norepinephrine may promote the release of anti-inflammatory cytokines, while activation of alfa-adrenergic receptors may enhance pro-inflammatory signaling [348]. This dual role of norepinephrine underlines the complexity of the relationship between the locus coeruleus, neuroinflammation, and mood regulation.

## 6. Conclusions

The present comprehensive review has explored the intricate relationship between neuroinflammation and depression, highlighting the complex interplay between the immune system, neurotransmitter function, and mood regulation. The evidence presented demonstrates that neuroinflammation plays a crucial role in the pathophysiology of depression, influencing multiple biological systems and contributing to the development and persistence of depressive symptoms.

Neuroinflammation, characterized by microglial activation and increased pro-inflammatory cytokine production, is a common feature in depression. This inflammatory state can disrupt neurotransmitter systems, impair neuroplasticity, and contribute to structural and functional changes in key brain regions involved in mood regulation. The HPA axis dysfunction observed in depression is closely linked to neuroinflammatory processes. Chronic stress and elevated cortisol levels can exacerbate inflammation, creating a vicious cycle that perpetuates depressive symptoms.

Peripheral immune system activation, particularly involving monocytes and macrophages, can contribute to central nervous system inflammation through various mechanisms, including blood–brain barrier disruption and activation of neuroinflammatory pathways.

The gut microbiota plays a significant role in modulating neuroinflammation and depression through the gut–brain axis. Alterations in gut microbial composition and metabolite production can influence neurotransmitter systems and immune function, potentially contributing to depressive symptoms.

Oxidative stress and mitochondrial dysfunction are closely associated with neuroinflammation in depression, further contributing to cellular damage and neurotransmitter imbalances.

The emerging evidence suggests that targeting neuroinflammation may offer new therapeutic approaches for depression, particularly in treatment-resistant cases or in patients with elevated inflammatory markers. The implications of these findings are far-reaching, suggesting a need for a paradigm shift in our understanding and treatment of depression. Moving forward, several key areas warrant further investigation:➢Development of more precise biomarkers for neuroinflammation in depression, which could help identify patients who may benefit from anti-inflammatory interventions;➢Exploration of novel anti-inflammatory agents and immunomodulatory therapies as potential treatments for depression, particularly in cases where traditional antidepressants have proven ineffective;➢Further elucidation of the mechanisms by which gut microbiota influence neuroinflammation and mood, potentially leading to probiotic or prebiotic interventions for depression;➢Investigation of personalized treatment approaches that consider an individual inflammatory profile, genetic predisposition, and environmental factors;➢Longitudinal studies to better understand the temporal relationship between neuroinflammation and depression, including potential preventive strategies targeting inflammatory processes.

Overall, the growing body of evidence linking neuroinflammation to depression underscores the need for a more holistic and integrative approach to understanding and treating this complex disorder. By considering the interplay between immune function, neurotransmitter systems, and environmental factors, we may be able to develop more effective and personalized interventions for individuals suffering from depression. As research in this field continues to evolve, it holds promise for improving outcomes and quality of life for millions of people affected by this debilitating condition.

## Figures and Tables

**Figure 1 biomolecules-15-00502-f001:**
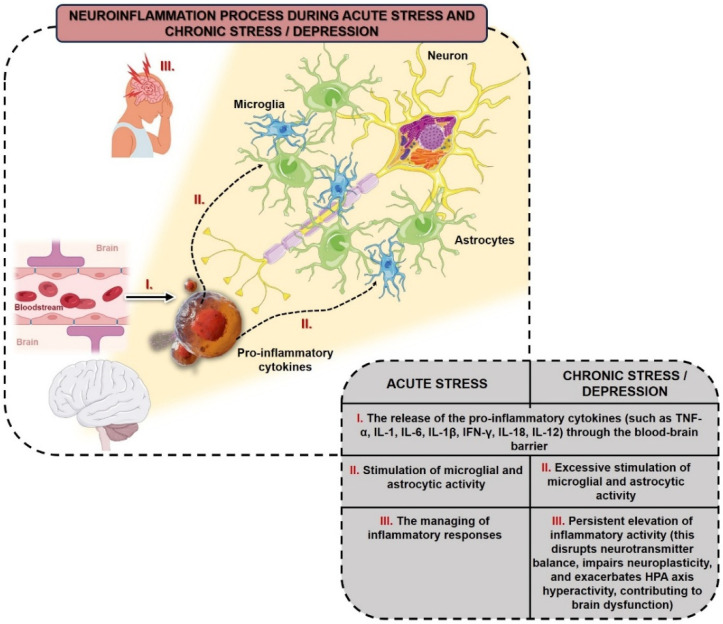
Neuroinflammatory responses in the brain during acute and chronic stress/depression. This model explains how chronic stress contributes to the pathophysiology of depression by promoting a proinflammatory state in the brain, suggesting that targeting inflammation may be beneficial in treating stress-related mental disorders.

**Figure 2 biomolecules-15-00502-f002:**
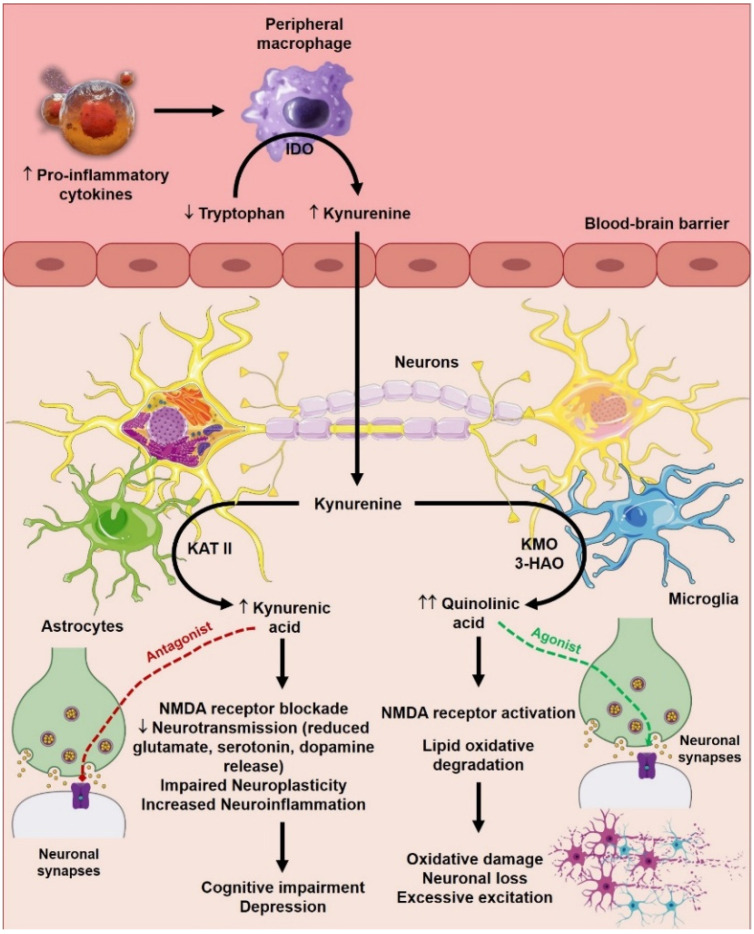
The role of the kynurenine pathway in neuroinflammation and depression: impact of proinflammatory cytokines on neurotransmitter balance and cognitive dysfunction. Proinflammatory cytokines and IDO activation: proinflammatory cytokines (IL-6, TNF-α) stimulate IDO activity in peripheral macrophages. IDO catalyzes the conversion of tryptophan to kynurenine, depleting tryptophan, which is essential for serotonin synthesis. This reduction in serotonin is associated with depressive symptoms. Transport and metabolism of kynurenine: kynurenine crosses the blood–brain barrier, where it is metabolized by astrocytes and microglia. In astrocytes, kynurenine is converted to kynurenic acid by kynurenine aminotransferase II (KAT II). Kynurenic acid acts as an antagonist at α7-nicotinic acetylcholine receptors (α7nAChRs), reducing the release of dopamine, serotonin, and glutamate. This leads to cognitive dysfunction. Quinolinic acid and neurotoxicity: In microglia, kynurenine is metabolized by kynurenine 3-monooxygenase (KMO) and 3-hydroxy anthranilic acid oxidase (3-HAO) to quinolinic acid. Quinolinic acid is a potent NMDA receptor agonist, leading to excitotoxicity and lipid peroxidation. These processes contribute to oxidative stress, neurodegeneration, and depressive symptoms.

**Figure 3 biomolecules-15-00502-f003:**
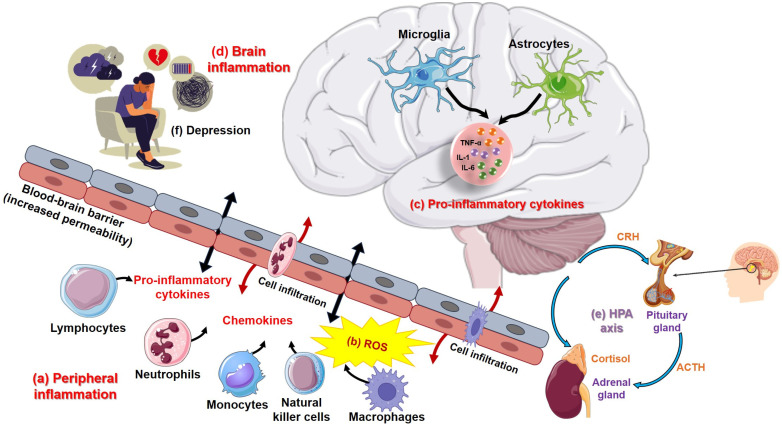
Interaction between peripheral and central inflammation, brain inflammation, and the HPA axis in depression. (a) Peripheral inflammation: peripheral immune cells, such as monocytes and lymphocytes, release pro-inflammatory cytokines (e.g., IL-1, IL-6, TNF-α) and chemokines in response to inflammation. These cytokines can enter the bloodstream, signaling inflammation throughout the body. (b) ROS: Excessive inflammation leads to the generation of ROS, which are highly reactive molecules that damage cells and amplify inflammatory signaling. (c) Communication to the brain: Cytokines from peripheral inflammation can pass through the BBB or signal through the endothelial cells of the BBB, triggering inflammation in the brain. This process activates microglia and astrocytes (key immune and support cells in the brain), which then release their proinflammatory cytokines, such as IL-1, IL-6, and TNF-α. (d) Brain inflammation: chronic brain inflammation contributes to neurological disorders, cognitive deficits, and behavioral changes such as anxiety or depression. The inflamed brain, as shown in the figure, is represented by activated microglia and astrocytes. (e) Impact on the HPA axis: Inflammation influences the HPA axis: Stress or inflammatory signals stimulate the hypothalamus to release CRH again. CRH triggers the pituitary gland to secrete ACTH. ACTH acts on the adrenal glands, causing the release of the stress hormone cortisol. Cortisol usually has anti-inflammatory effects, but prolonged activation (chronic stress) can dysregulate this system, leading to persistent inflammation and impaired stress responses. (f) Psychological effects: Stress and inflammation are described as affecting mental health and can cause depressive symptoms and brain dysfunction.

**Figure 4 biomolecules-15-00502-f004:**
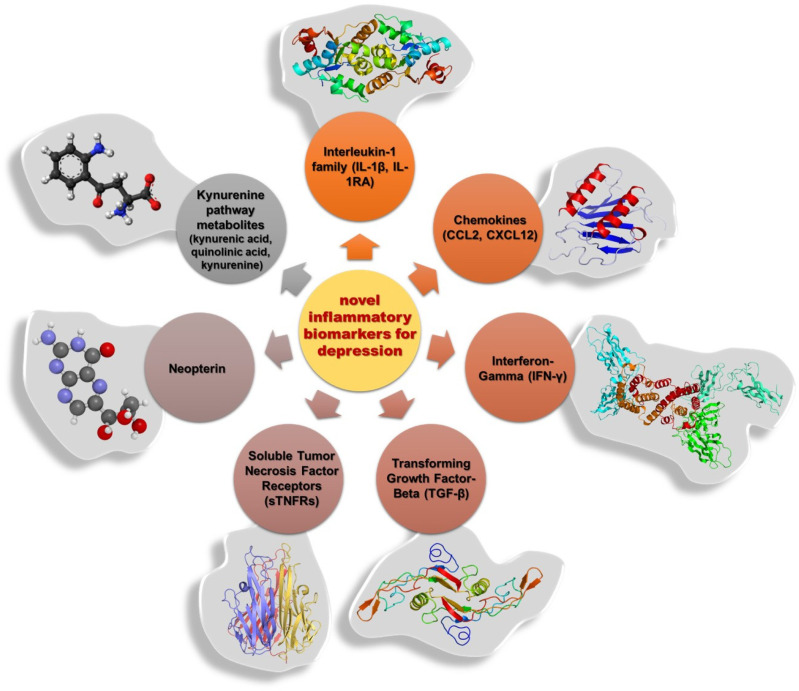
Novel inflammatory biomarkers for depression.

**Figure 5 biomolecules-15-00502-f005:**
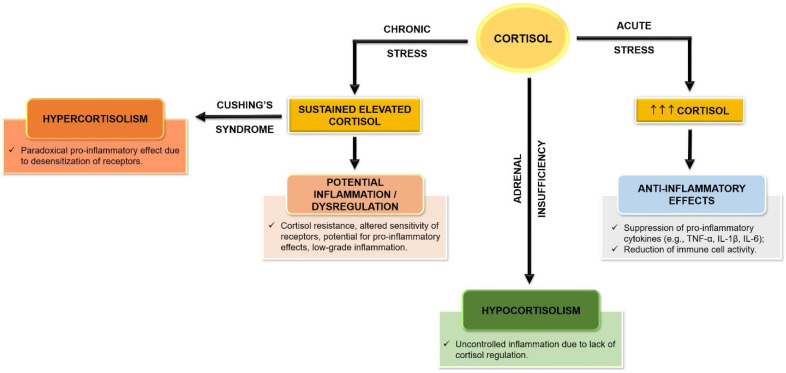
The complex relationship between cortisol and inflammation.

**Table 1 biomolecules-15-00502-t001:** Randomized clinical trials of anti-inflammatory drugs in depression.

Type of Study	Subjects Included in the Study	Treatment	Key Findings	References
Randomized Clinical Trial	Minocycline group (n = 24), Placebo group (n = 26); Patients with treatment-resistant depression	Minocycline (200 mg daily) vs. Placebo	Minocycline significantly improved depressive symptoms compared to placebo.	[277]
Randomized, Double-Blind, Placebo-Controlled Study	Healthy adults experiencing stress and anxiety (n = 54)	Ashwagandha root extract (500 mg daily) vs. Placebo	Ashwagandha reduced stress and anxiety, increased serotonin, and lowered cortisol levels.	[278]
Randomized Controlled Trial	Obese patients with type 2 diabetes and depression (n = 227)	Curcumin (dose not specified) vs. Placebo	Curcumin improved depression severity and reduced inflammatory markers in patients with type 2 diabetes.	[279]
Randomized, Double-Blind, Placebo-Controlled Trial	Patients with moderate to severe depression (n = 90)	Empagliflozin (10 mg daily) + Citalopram vs. Citalopram + Placebo	Empagliflozin significantly reduced depression severity over time compared to placebo.	[280]
Randomized Clinical Trial	Young adults with moderate to severe MDD (n = 130)	Aspirin (100 mg daily), Rosuvastatin (10 mg daily) vs. Placebo	Neither aspirin nor rosuvastatin showed significant improvement in depression; rosuvastatin showed some secondary benefits.	[281]
Randomized Clinical Trial	Older adults (n = 19,114)	Aspirin (100 mg daily) vs. Placebo	Aspirin did not reduce the risk of depression in older adults.	[282]
Randomized Controlled Trial	Patients with high and low inflammation MDD (n = 200)	Vortioxetine + Celecoxib vs. Vortioxetine + Placebo	Vortioxetine + Celecoxib showed potential for greater improvement in inflammation-related depression.	[283]
Randomized, Double-Blind, Placebo-Controlled Study	Patients with major depressive disorder (n = 48)	Low-dose Naltrexone vs. Placebo	Naltrexone improved depressive symptoms, particularly in those with elevated inflammation markers.	[284]
Randomized, Double-Blind, Placebo-Controlled Trial	TRD patients with increased inflammatory activity (n = unknown)	N-acetylcysteine + Antidepressants vs. Placebo + Antidepressants	N-acetylcysteine showed promise in TRD patients with inflammation, improving depression symptoms.	[285]
Randomized Placebo-Controlled Clinical Trial	Patients with SSRI treatment (n = 56)	Glycyrrhizic Acid (GZA) + SSRI vs. SSRI + Placebo	GZA significantly improved depressive symptoms, with better response in patients with higher baseline inflammation.	[286]

## Data Availability

No new data were created.

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
