# Peer review of "Neuroinflammation—A Crucial Factor in the Pathophysiology of Depression—A Comprehensive Review"

_biomolecules, 2025, doi:10.3390/biom15040502_

Round 1
Reviewer 1 Report
Comments and Suggestions for Authors
The presented review should be structured. It is necessary to immediately identify the signaling pathways involved in the formation of depression and then strictly adhere to the plan. In its current form, the article looks chaotic. For example, it is about the Nf-kB signaling pathway (2.4. NF-κB Signaling Pathway in Neuroinflammation), but then comes section 2.5. Blood-Brain Barrier Dysfunction. These are different levels of regulation. It is very difficult to follow the logic of the review article with this approach to writing it. Given the huge amount of text information, the number of figures seems insufficient, despite their good quality and content.
1) Abbreviations in section titles should be avoided
2) Figures begin with section 2.2. Role of Cortisol and Stress Hormones in Inflammation. And the next figure is cited on page 16. At the same time, a lot of text information is presented. The authors should think about creating tables and additional figures.
Comments on the Quality of English Language
The text of the article is very extensive. Contains a fairly large number of grammatical and punctuation errors.
Author Response
Dear reviewer ,
Please find attached the answer to the suggestions you have addressed to us. Thank you for your cooperation.

Reviewer 2 Report
Comments and Suggestions for Authors
Sălcudean et al. present a broad and comprehensive narrative review of the neuroinflammatory theory of depression. They incorporate the relationship with the HPA axis, of primary interest in understanding the pathophysiology of this complex disorder. I could make only two suggestions:
- Briefly discuss the relationship between childhood maltreatment, inflammation, and depression. There is abundant evidence in this regard, including gene-environment interaction models.
- In a final section, briefly discuss the clinical evidence on the use of anti-inflammatory drugs in depression. Please cite randomized clinical trials in humans or systematic reviews with meta-analysis.
Author Response

(The authors gave the same response as above.)

Reviewer 3 Report
Comments and Suggestions for Authors
The manuscript from Sălcudean et all, provides a comprehensive review of the role of neuroinflammation in the pathophysiology of depression. It effectively integrates current research, covering various biological mechanisms, including cytokine signaling, HPA axis dysregulation, neurotransmitter imbalances, and the blood-brain barrier dysfunction. The review is supported by relevant citations and provides valuable insights into the interplay between immune responses and mood disorders. However, certain areas require improvement to enhance clarity, coherence, and impact. The organization of topics could be more structured to facilitate readability, and some sections would benefit from greater clarity and conciseness. Additionally, there are instances of redundancy, and the integration of figures could be more precise to better illustrate key concepts in conveying the complex relationship between neuroinflammation and depression.
Major:
- The current layout is confusing and occasionally repetitive, making it difficult to follow. To enhance clarity, I suggest structuring the discussion of depression across different diseases more cohesively and consolidating related topics into a logical and sequential format.
- Since the focus of this revision is the impact of neuroinflammation on neurotransmitter systems in depression, I would like to suggest that the authors start by discussing the definition of neuroinflammation and its global status in depression, as well as the neurotransmitter system, to contextualize the topic, the problem, and the challenges.
- Several important terms are missing and should be defined to contextualize the discussion, including “acute stress,” “chronic stress,” “Cushing’s syndrome,” “cognitive-behavioral therapy (CBT),” and “mindfulness-based stress reduction.
- Microglia’s Expanded Role: Microglia are not limited to immune functions; they are critical regulators of neuronal activity through synaptic remodeling. Furthermore, microglia-astrocyte and microglia-oligodendrocyte crosstalk significantly influence microglial transcriptomic signatures. The authors should discuss how these mechanisms contribute to depression.
- Line 462: The discussion on macrophages in depression should be more precise. Under homeostatic conditions, microglia and CNS-associated macrophages (leptomeningeal, perivascular, and choroid plexus macrophages) are distinct based on niche localization and marker expression. However, in disease states, spatial reorganization and transcriptomic changes “mask” these differences, making it difficult to determine which immune cells contribute most to depression onset and progression. A more detailed discussion would strengthen this section.
- Are the authors considering the BBB’s role only in depression linked to neurodegenerative disorders? What about its role in mood disorders more broadly? A discussion on its involvement in non-neurodegenerative depression would provide a more comprehensive perspective.
- To enhance the clarity and impact of the manuscript, I suggest structuring the discussion of depression across different diseases more cohesively. The current layout is somewhat confusing and occasionally repetitive, which affects readability. A more streamlined approach would improve the manuscript’s flow and make the content more engaging and comprehensible.
- A brief introduction on the main brain regions affected in depression should be added before digging into specific areas. Additionally, the authors should clarify why they focus exclusively on the nucleus accumbens. Given that depression involves widespread neural circuit dysfunction (including the prefrontal cortex, amygdala, hippocampus, anterior cingulate cortex, and insular cortex), expanding the discussion or providing a justification for this focus would enhance clarity.
- The authors should organize the transition topics more clearly, as the transition between sections is abrupt. For instance, the discussion on neuroinflammation's impact on mood disorders appears suddenly, making it difficult for the reader to follow the manuscript's logical flow.
- I recommend including a table summarizing key mediators involved in depression, such as cytokines, hormones, receptors, reactive oxygen species (ROS), and neurotransmitters. This would provide a clearer and more structured overview of the key factors influencing depression.
- Lines 746-749: How do microglia contribute to these processes?
- Line 1401: The M1/M2 macrophage dichotomy is no longer a valid classification, as macrophage polarization exists on a spectrum of phenotypic states rather than a binary model. I recommend that the authors update their discussion to reflect the current understanding of macrophage/microglia plasticity, considering recent literature on context-dependent functional states and the influence of the microenvironment on macrophage behavior.
- The authors should expand their discussion on how systemic immune responses, including immune cell infiltration and cytokine signaling (e.g., regulatory T/B cells, dendritic cells), contribute to CNS neuroinflammation.
- The paper relies mostly on cross-sectional studies but does not critically assess the lack of longitudinal data. The authors should emphasize the need for studies tracking neuroinflammation over time in depression patients to determine causality and progression.
- The manuscript presents evidence for inflammation-induced depression but does not discuss how depression itself might contribute to neuroinflammation. I encourage the authors to add a section discussing where chronic depression alters immune function, increasing pro-inflammatory markers, etc.
- The manuscript mentions IL-6, TNF-α, and CRP as inflammatory markers but does not explore novel or more specific biomarkers for depression. I encourage the authors to discuss emerging biomarkers, such as metabolites, lipid-based markers, or neuroimmune signatures, for stratifying patients for personalized treatment approaches.
- The gut microbiota significantly influences neuroinflammation, neurotransmitter production, and HPA axis regulation, all of which are implicated in depression. I am surprised that the authors did not discuss how gut dysbiosis, microbial-derived metabolites (e.g., short-chain fatty acids), and gut permeability alterations contribute to depressive symptoms and inflammation-driven mood disorders.
- One major limitation of the manuscript is the lack of discussion of the role of emerging technologies in studying depression and neuroinflammation (e.g., neuroimaging techniques; AI and machine learning for predictive models of inflammation-driven depression, biomarker-based patient stratification; microfluidics and organ-on-chip models; single-cell sequencing and transcriptomics).
Minor:
- Some sentences are long and complex, making comprehension difficult. Figure 1 needs a description. Be consistent with figure descriptions when using (or not) sentence case for figures titles.
- Figures 2 -4: Both microglia and astrocytes morphologies do not correspond to an activated state (according to the author's description in the manuscript).
- The authors should define key terms upon first use and ensure uniform terminology throughout the manuscript. For instance, the manuscript fluctuates between “glial activation,” “microglial response,” and “neuroimmune activation.”
- The manuscript sometimes shifts between technical and general descriptions, which may reduce coherence. Consistently using precise terminology will improve readability.
- The authors must ensure that all references are properly cited and that citation formatting is consistent. For instance: "Recent studies have highlighted..." without a direct reference. Also, some references are incomplete.
- Some minor grammatical inconsistencies need revision to enhance clarity.
Author Response

(The authors gave the same response as above.)

Round 2
Reviewer 1 Report
Comments and Suggestions for Authors
The article can be accepted in its current form.
Author Response
Thank you very much for your time and your kind appreciation.
The Authors
Reviewer 3 Report
Comments and Suggestions for Authors
Dear authors,
Thank you for your thorough and thoughtful revisions. The manuscript has significantly improved in both clarity and structure, and I have no further major comments.
I would only like to highlight a minor editorial point to address before the final submission:
Several abbreviations have been redefined multiple times throughout the manuscript, particularly when introducing new sections. The same applies to the definition of neuroinflammation.
Please ensure that each abbreviation is defined only once at its first mention and used consistently thereafter. Likewise, consider maintaining a single, clear definition of neuroinflammation to enhance consistency and readability.
Once these minor points are addressed, the manuscript will be ready for publication.
Best regards,
Warm regards
Author Response
Thank you very much for your time. Please find attached our response.
The authors.
